# Understanding Deep Neural Function Approximation in Reinforcement Learning via $\epsilon$-Greedy Exploration

**Fanghui Liu,**[*] **Luca Viano, Volkan Cevher**
Laboratory for Information and Inference Systems
École Polytechnique Fédérale de Lausanne (EPFL), Switzerland
`{first}.{last}@epfl.ch`

## Abstract

This paper provides a theoretical study of deep neural function approximation in reinforcement learning (RL) with the $\epsilon$-greedy exploration under the online setting. This problem setting is motivated by the successful deep Q-networks (DQN) framework that falls in this regime. In this work, we provide an initial attempt on theoretical understanding deep RL from the perspective of function class and neural networks architectures (e.g., width and depth) beyond the "linear" regime. To be specific, we focus on the value based algorithm with the $\epsilon$-greedy exploration via deep (and two-layer) neural networks endowed by Besov (and Barron) function spaces, respectively, which aims at approximating an $\alpha$-smooth Q-function in a $d$-dimensional feature space. We prove that, with $T$ episodes, scaling the width $m = \widetilde{\mathcal{O}}(T^{\frac{d}{2\alpha+d}})$ and the depth $L = \mathcal{O}(\log T)$ of the neural network for deep RL is sufficient for learning with sublinear regret in Besov spaces. Moreover, for a two layer neural network endowed by the Barron space, scaling the width $\Omega(\sqrt{T})$ is sufficient. To achieve this, the key issue in our analysis is how to estimate the temporal difference error under deep neural function approximation as the $\epsilon$-greedy exploration is not enough to ensure "optimism". Our analysis reformulates the temporal difference error in an $L^2(\mathrm{d}\mu)$-integrable space over a certain averaged measure $\mu$, and transforms it to a generalization problem under the non-iid setting. This might have its own interest in RL theory for better understanding $\epsilon$-greedy exploration in deep RL.

## 1 Introduction

Efficient reinforcement learning (RL) under the large (or even infinite) state space and action space setting is increasingly important and relevant challenge [1, 2, 3]. One of the first successful approaches towards this problem is the deep Q-network (DQN) [4, 5] framework, which deploys powerful nonlinear function approximation techniques via Deep Neural Networks (DNNs) [6] to concisely approximate state and action spaces. Despite its impressive practical success, there is still a gap between practical uses and theoretical understanding on deep RL with regard to the function class and the employed $\epsilon$-greedy policy.

In the perspective of function class, many theoretical works center around linear function approximation [7, 8] and linear mixtures [9, 10]. Existing non-linear function approximation results on RL are largely based on neural tangent kernel (NTK) [11, 12], Bellman rank [13, 14], and Eluder dimension [15, 16, 17]. Nevertheless, these approaches fail in truly capturing the highly non-linear properties of deep RL. For example, NTK (or lazy training [18]) essentially works in a "linear" regime [19, 20, 21], and can not efficiently learn even a single ReLU neuron [22, 23, 24] as it requires $\Omega(\varepsilon^{-d})$ samples

---

[*]Correspondence to: Fanghui Liu <`fanghui.liu@epfl.ch`> and Luca Viano<`luca.viano@epfl.ch`>.

36th Conference on Neural Information Processing Systems (NeurIPS 2022).

to achieve $\varepsilon$ approximation error, where $d$ is the (original) or transformed feature dimension input; the Bellman rank is normally difficult to be estimated for neural networks as suggested by [25]; the Eluder dimension is at least in an exponential order [26, 27] even for two-layer neural networks. The above general function approximation schemes appear difficult to fully demonstrate the success of practical deep RL both theoretically and empirically.

In the perspective of exploration schemes, DQN is directly equipped with the $\epsilon$-greedy policy instead of confidence-bound based scheme that are commonly used in RL theory. The $\epsilon$-greedy exploration is theoretically demonstrated to have exponential sample complexity in the worst case [28] but is still popular in practical deep RL due to its simple implementation. In this case, theoretical analyses of $\epsilon$-greedy in deep RL are still required. Besides, to ensure a sublinear regret, under the NTK regime, the width of neural networks is required to be $m = \Omega(T^{13})$ [12], where $T$ is the number of episodes. This does not match deep RL in practice with small width/depth under large episodes [4, 29].

To bridge the large theory-practice gap, we study the value iteration algorithm with deep neural function approximation and the $\epsilon$-greedy policy under the online setting, which broadly captures the key features of DQN. Our analysis framework is based on DNNs (as well as two-layer neural networks) where the target Q function lies in the Besov space [30] or the Barron space [31], respectively. These function classes can fully capture the properties of Q-functions, e.g., smoothness by neural networks. Our results demonstrate that the sublinear regret can be achieved for deep neural function approximation under the $\epsilon$-greedy exploration with reasonably finite width and depth in practice. Besides, the relationship between the problem-dependent smoothness of Q-function and regret bounds is also developed. These results could also motivate practitioners to consider different architectures of implementations of deep RL.

## 1.1 Technical challenges and contributions

Most previous RL theory results on function approximation in the online setting work with "optimism in the face of uncertainty" principle for exploration, leading to a series of upper confidence bound (UCB)-type algorithms to ensure the temporal difference (TD) error smaller than zero.

Conceptually, optimism is sometimes too aggressive and UCB-style algorithms can suffer exponential sample complexity even for nonlinear bandits [27]. Technically, UCB-type algorithms in linear/kernel function approximation [7, 12, 32] depend on a known feature mapping or the NTK kernel, which appears invalid for deep neural function approximation beyond the "linear" regime. This is because, the used confidence ellipsoid and elliptical potential lemma are not applicable for data-dependent feature mapping of DNNs. To avoid explicitly designing a bonus function, Thompson sampling [33, 34] appears promising in a Bayesian perspective by using randomized (i.e., perturbed) versions of the estimated model or value function [35]. Nevertheless, the bonus function is still implicitly included in confidence estimate of perturbations.

In this work, we center around deep neural function approximation with the $\epsilon$-greedy exploration. Since this exploration scheme is not enough to ensure the TD error smaller than zero, the technical challenge in our analysis is how to estimate it to ensure the sublinear regret. In our proof framework, by a measure transform, the TD error is analysed in an $L^2(\mathrm{d}\bar{\mu})$-integrable space, where $\bar{\mu}$ is the averaged measure wrt a mini-batch of historical state-action pairs. To break the dependence between the episodes for neural networks training, we utilize the *experience replay* scheme [36] from DQN, and then transform the TD error estimation to generalization error under the independent but non-identically distributed data setting and approximation error in the respective function spaces. Note that in practice, experience replay makes observations to be (nearly) iid, but our analysis only requires the independence of observations, that is weaker than iid. Such generalization problem can be addressed by uniform convergence via (local) Rademacher complexity of the Besov/Barron spaces under the averaged measure. This considered function spaces in this work is more general than Hölder spaces used in offline RL [37].

Our results show that (*i*) the problem-dependent smoothness of Q-function affects the efficiency of learning with deep RL, which can be improved by increasing the model capacity (width and depth). We use $\alpha$ as a parameter indicating the smoothness degree of Q-function. A larger $\alpha$ indicates smoother functions, easier RL tasks, and smaller exploration times, which coincides with our theory. (*ii*) for deep neural networks under the Besov space, the width $m = \widetilde{\mathcal{O}}(T^{\frac{d}{2\alpha+d}})$ and the depth $L = \widetilde{\mathcal{O}}(1)$ are enough for sublinear regret under the $\epsilon$-greedy policy, where $\widetilde{\mathcal{O}}(\cdot)$ omits the log

terms. (*iii*) for two-layer neural networks under the Barron space, the width $m = \Omega(\sqrt{T})$ suffices to ensure sublinear regret. Furthermore, our regret bounds can be independent of the feature dimension, supporting the premise of practical, high-dimensional data in RL.

## 1.2 Related work

Recent work on neural network function approximation beyond NTK (or the Eluder dimension) mainly restrict on the generative setting [25, 38] by assuming a simulator in which the agent can require any state and action, and the offline setting [37, 39]. In sequel, we review RL with function approximation under the online setting that DQN falls into this regime. We also mention that, theoretical understanding of DQN can be conducted by from the perspective of neural fitted Q-iteration algorithm [40, 37, 41], and Q learning [42] in the perspective of understanding the target network [43] and experience replay [44, 45, 46] with linear function approximation. Note that, for notational consistency with previous work, in this subsection, $T$ denotes the total number of steps (i.e., interactions with the environment) instead of the number of episodes in our paper.

**RL with linear/kernel function approximation:** RL with linear function approximation achieves a sublinear regret bound with $\widetilde{\mathcal{O}}(\sqrt{d^3 H^3 T})$ under a low-rank MDP in a model-free setting [7] and $\widetilde{\mathcal{O}}(dH^2\sqrt{T})$ in a model-based setting [32], where $H$ is the length of each episode. The regret can be improved to $\widetilde{\mathcal{O}}(dH\sqrt{T})$ under a low inherent Bellman error by assuming a global planning oracle [47] or under a Bernstein-type exploration bonus and controlling extra uniform convergence cost [48]. This nearly optimal regret can be also achieved under the linear mixtures setting [10]. In the kernel regime, the regret can be achieved with $\widetilde{\mathcal{O}}(\delta_{\mathcal{F}}\sqrt{H^3 T})$ [32, 12], where $\delta_{\mathcal{F}}$ is the intrinsic complexity (e.g., effective dimension) of the function class RKHS $\mathcal{F}$. The above bounds are based on confidence ellipsoid to quantify the uncertainty in an explicit bonus function by feature mapping/kernel function; while Thompson sampling [34, 33] utilizes an implicit bonus function in probability estimation on uncertainty quantification, which leads to an $\widetilde{\mathcal{O}}(d^2 H^2\sqrt{T})$ [35] regret in linear function approximation.

**RL with general function approximation:** One prototypical scheme uses the Eluder dimension [15], which measures the degree of dependence among action rewards, resulting in an $\widetilde{\mathcal{O}}(\texttt{poly}(\delta_{\mathcal{F}} H)\sqrt{T})$ regret [16, 17], where the complexity $\delta_{\mathcal{F}}$ depends on the Eluder dimension. Using this metric, the sublinear regret under the $\epsilon$-greedy exploration can be achieved by [49]. Besides, the low Bellman rank assumption [13], where the Bellman error "matrix" admits a low-rank factorization, can be also used general function approximation [14] by measuring the error of the function class under the Bellman operator. Combining Bellman rank and Eluder dimension results in a new metric, Bellman Eluder dimension [50], achieving $\widetilde{\mathcal{O}}(H\sqrt{\delta_{\mathcal{F}} T})$-regret, where $\delta_{\mathcal{F}}$ depends on this metric.

Overall, the above metrics are difficult to the nonlinear spaces of DNNs beyond "linear" regime that concern us.

## 2 Background and preliminaries

In this section, we introduce the necessary background and definitions with respect to online reinforcement learning based on episodic Markov decision processes (MDPs) and function spaces of deep (and two-layer) ReLU neural networks.

**Notation:** We denote by $a(n) \lesssim b(n)$: there exists a positive constant $c$ independent of $n$ such that $a(n) \leqslant cb(n)$; $a(n) \asymp b(n)$: there exists two positive constant $c_1$ and $c_2$ independent of $n$ such that $c_1 b(n) \leqslant a(n) \leqslant c_2 b(n)$. We use the shorthand $[n] := \{1, 2, \ldots, n\}$ for some positive $n$ and $\lceil x \rceil$ denotes the smallest integer exceeding $x$. Let $\mathcal{X} = [0,1]^d$ be a domain of the functions, we denote the $L^p$-integrable space by $L^p(\mathcal{X})$ endowed by the norm $\|f\|_{L^p(\mathcal{X})} = \left( \int_{\mathcal{X}} |f(\boldsymbol{x})|^p \mathrm{d}\boldsymbol{x} \right)^{1/p}$, and the $\mu$-integrable $L^p$ space by $L^p(\mathrm{d}\mu)$ for a probability measure $\mu$ on $\mathcal{X}$ and the norm is given by $\|f\|_{L^p(\mathrm{d}\mu)} = \left( \int_{\mathcal{X}} |f(\boldsymbol{x})|^p \mathrm{d}\mu \right)^{1/p}$.

### 2.1 Episodic Markov decision processes

A (finite-horizon) episodic MDPs is denoted as MDP$(\mathcal{S}, \mathcal{A}, H, \mathbb{P}, r)$, where $\mathcal{S}$ is the state space with possibly infinite states; $\mathcal{A}$ is the finite action space; $H$ is the number of steps in each episode;

$\mathbb{P} := \{\mathbb{P}_h\}_{h=1}^H$ is the Markov transition kernel with the transition probability $\mathbb{P}_h(\cdot|s, a)$ on action $a$ taken at state $s \in \mathcal{S}$ in the $h$-th step; the reward functions $r := \{r_h\}_{h=1}^H$ are assumed to be deterministic. For notational simplicity, denote $\mathcal{X} = \mathcal{S} \times \mathcal{A}$ and $\boldsymbol{x} = (s, a)$, we assume $\mathcal{X} = [0, 1]^d$ as a compact space of $\mathbb{R}^d$ and $r_h : \mathcal{S} \times \mathcal{A} \to [0, 1]$ at $h$-th step.

A non-stationary policy $\pi$ is a collection of $H$ functions $\pi := \{\pi_h : \mathcal{S} \to \mathcal{A}\}_{h=1}^H$. Given a policy $\pi$, the (state) value function $V_h^\pi : \mathcal{S} \to [0, H]$ is defined as the expected cumulative reward of the MDP starting from step $h \in [H]$, i.e., $V_h^\pi(s) = \mathbb{E}_\pi\left[ \sum_{h'=h}^H r_{h'}(s_{h'}, a_{h'}) \big| s_h = s \right], \forall s \in \mathcal{S}, h \in [H]$ where $\mathbb{E}_\pi[\cdot]$ denotes the expectation with respect to the randomness of the trajectory $\{(s_h, a_h)\}_{h=1}^H$ obtained by the policy $\pi$. Likewise, the action-value function $Q_h^\pi : \mathcal{S} \times \mathcal{A} \to [0, H]$ is defined as $Q_h^\pi(s, a) = \mathbb{E}_\pi\left[ \sum_{h'=h}^H r_{h'}(s_{h'}, a_{h'}) \big| s_h = s, a_h = a \right]$.

Moreover, since the action space and episode length are both finite, there always exists an optimal policy $\pi^\star$ [51] such that $V_h^\star(s) = \sup_\pi V_h^\pi(s)$ for all $s \in \mathcal{S}$ and $h \in [H]$. To simplify the notation, denote $(\mathbb{P}_h V)(s, a) := \mathbb{E}_{s' \sim \mathbb{P}_h(\cdot|s,a)}[V(s')]$ and the Bellman operator $(\mathbb{T}_h V)(s, a) = r_h(s, a) + (\mathbb{P}_h V)(s, a)$ for any measurable function $V : \mathcal{S} \to [0, H]$. Using this notation, the Bellman equation associated with a policy $\pi$ can be formulated as

$$Q_h^\pi(s, a) = (\mathbb{T}_h V_{h+1}^\pi)(s, a), \qquad V_h^\pi(s) = \langle Q_h^\pi(s, \cdot), \pi_h(\cdot|s) \rangle_\mathcal{A}, \qquad V_{H+1}^\pi(s) = 0. \qquad (1)$$

Similarly, the Bellman optimality equation is given by

$$Q_h^\star(s, a) = (\mathbb{T}_h V_{h+1}^\star)(s, a), \qquad V_h^\star(s) = \max_{a \in \mathcal{A}} Q_h^\star(s, a), \qquad V_{H+1}^\star(s) = 0. \qquad (2)$$

Accordingly, the optimal policy $\pi^\star$ is the greedy policy with respect to $\{Q_h^\star\}_{h=1}^H$. Hence the Bellman optimality operator $\mathbb{T}_h^\star$ is defined as

$$(\mathbb{T}_h^\star Q)(s_h, a_h) = r_h(s_h, a_h) + \mathbb{E}_{s_{h+1} \sim \mathbb{P}_h(\cdot|s_h, a_h)}[\max_{a \in \mathcal{A}} Q(s_{h+1}, a)], \qquad \forall Q : \mathcal{S} \times \mathcal{A} \to [0, H].$$

By definition, the Bellman equation in Eq. (2) is equivalent to $Q_h^\star = \mathbb{T}_h^\star Q_{h+1}^\star, \forall h \in [H]$.

In the **online setting**, the goal is to learn the optimal policy $\pi^\star$ by minimizing the cumulative regret under the interaction with the environment over a number of episodes. For any policy $\pi$, the difference between $V_1^\pi$ and $V_1^\star$ quantifies its sub-optimality. Thus, after $T$ (fixed but large) episodes, the total (expected) regret is defined as $\text{Regret}(T) = \sum_{t=1}^T \left[ V_1^\star(s_1^t) - V_1^{\tilde{\pi}^t}(s_1^t) \right]$, where $\tilde{\pi}^t$ is the policy executed in the $t$-th episode and $s_1^t$ is the initial state.

## 2.2 Function spaces

We give an overview of Besov spaces for deep neural networks and the Barron space for two-layer neural networks. More details refer to Appendix A. For description simplicity, we focus on the ReLU activation function in this work.

**Besov spaces:** Previous work in approximation theory focuses on the "smoothness" of the function, e.g., Hölder spaces [52, 37] and Sobolev spaces [53]. Here we consider the concept of $\alpha$-smooth from modulus of smoothness [30], *cf.*, Appendix A.

Based on this, we consider a more general function space beyond Hölder spaces and Sobolev spaces, i.e., Besov spaces [54, 30], which allows for spatially inhomogeneous smoothness with spikes and jumps. The Besov space is defined by $\mathcal{B}_{p,q}^\alpha(\mathcal{X}) = \{f \in L^p(\mathcal{X}) \mid \|f\|_{\mathcal{B}_{p,q}^\alpha} < \infty\}$, where the Besov norm is $\|f\|_{\mathcal{B}_{p,q}^\alpha} := \|f\|_{L^p(\mathcal{X})} + |f|_{\mathcal{B}_{p,q}^\alpha}$. The smoothness parameter $\alpha$ indicates which function at a certain smoothness degree can be represented. For example, if $\alpha > d/p$, then the related Besov space is continuously embedded in the set of the continuous functions; if $\alpha < d/p$, then the functions in the Besov space are no longer continuous. The formal definition and relations to Hölder spaces and Sobolev spaces are deferred to Appendix A.

**Barron spaces:** A two-layer neural network with $m$ neurons can be represented as $f(\boldsymbol{x}) = \frac{1}{m} \sum_{k=1}^m b_k \sigma(\boldsymbol{w}_k^\top \boldsymbol{x} + c_k)$ with the ReLU activation function $\sigma(\cdot)$ used in this work and the neural network parameters $\{b_k, \boldsymbol{w}_k, c_k\}_{k=1}^m$. It admits the integral representation $f(\boldsymbol{x}) = \int_\Omega b\sigma\left(\boldsymbol{w}^\top \boldsymbol{x} + c\right) \rho(\mathrm{d}b, \mathrm{d}\boldsymbol{w}, \mathrm{d}c), \boldsymbol{x} \in \mathcal{X}$, where $\Omega = \mathbb{R} \times \mathbb{R}^d \times \mathbb{R}$ and $\rho$ is a probability measure over $\Omega$. Then the Barron space [31] endowed by the Barron norm is defined as

$$\widetilde{\mathcal{P}} = \left\{ f \text{ admits Eq. (12)} : \|f\|_{\widetilde{\mathcal{P}}} = \inf_\rho \{\mathbb{E}_\rho |b|(\|\boldsymbol{w}\|_1 + |c|)\} < \infty \right\}.$$

The Barron space $\widetilde{\mathcal{P}}$ [31] can be (roughly) equipped with the $\ell_1$-path norm, i.e., $\|f\|_{\widetilde{\mathcal{P}}} \leqslant \|f\|_{\mathcal{P}} := \frac{1}{m} \sum_{k=1}^{m} |b_k| (\|\boldsymbol{w}_k\|_1 + c_k) \leqslant 2\|f\|_{\widetilde{\mathcal{P}}}$. Accordingly, it is natural to use $\|f\|_{\mathcal{P}}$ to denote the Barron norm, as the discrete version.

The Barron space [31] can be regarded as the *largest* function space for two-layer ReLU neural networks. Here the "largest" terminology [31, 55] means that the approximation ability can avoid *curse of dimensionality*, i.e., 1) any function in Barron spaces can be efficiently approximated by two-layer neural networks with bounded norm; 2) any continuous function that can be efficiently approximated by two-layer neural networks with bounded norm belongs to a Barron space.

We remark that, avoiding curse of dimensionality is important in theory for practical high-dimensional data in RL. However, Besov spaces are too large and thus do not enjoy this property for deep ReLU neural networks.

## 3 Algorithm: Value iteration via DNNs under $\epsilon$-greedy exploration

In this section, we lay out our algorithm 1 via value iteration by DNNs under the $\epsilon$-greedy policy. Though our value iteration algorithm is different from one gradient-step for deep Q-learning in DQN, it still shares the key spirit with DQN in terms of function approximation via DNNs, $\epsilon$-greedy exploration, and experience replay.

**Function class:** We define the function class $\mathcal{F}$ given by $\mathcal{F} = \mathcal{F}_1 \times \cdots \times \mathcal{F}_H$, including $\mathcal{F}_{\text{SNN}}$ for two-layer (Shallow) ReLU neural networks and $\mathcal{F}_{\text{DNN}}$ for deep ReLU neural networks as below

$$\mathcal{F}_{\text{SNN}} = \left\{ f : [0,1]^d \to [0,H] \Big| f(\boldsymbol{x}) = \frac{1}{m} \sum_{k=1}^{m} b_k \sigma(\boldsymbol{w}_k^\top \boldsymbol{x} + c_k), \|f\|_{\mathcal{P}} \leqslant B \right\}, \tag{3}$$

where $B > 0$ is the $\ell_1$-path norm constraint parameter, and deep ReLU neural networks [30] as

$$\mathcal{F}_{\text{DNN}}(L, m, S, B) := \left\{ f : [0,1]^d \to [0,H] \Big| f(\boldsymbol{x}) = (\boldsymbol{W}^{(L)} \sigma(\cdot) + b^{(L)}) \circ \cdots \circ (\boldsymbol{W}^{(1)} \boldsymbol{x} + \boldsymbol{b}^{(1)}), \right.$$

$$\left. \sum_{i=1}^{L} (\|\boldsymbol{W}^{(i)}\|_0 + \|\boldsymbol{b}^{(i)}\|_0) \leqslant S, \ \max_i (\|\boldsymbol{W}^{(i)}\|_\infty \vee \|\boldsymbol{b}^{(i)}\|_\infty) \leqslant B \right\}, \tag{4}$$

where the weight parameters are $\boldsymbol{W}^{(1)} \in \mathbb{R}^{m \times d}$, $\boldsymbol{W}^{(l)} \in \mathbb{R}^{m \times m}$, $\forall l \in \{2, 3, \ldots, L-1\}$, and $\boldsymbol{W}^{(L)} \in \mathbb{R}^m$; the bias parameter are $\boldsymbol{b}^{(l)} \in \mathbb{R}^m$, $\forall l \in [L-1]$ and $b^{(L)} \in \mathbb{R}$. Such sparsely-connected neural networks require most of the network parameters to be zero or non-active, which can be verified [56]. The depth $L$, the width $m$, the sparsity parameter $S$ and the norm parameter $B$ can be determined later in our proof to achieve good approximation and estimation performance.

**Experience replay:** In our setting, after initialization, at $t$-th episode, at $h$-th time step, we have observed $t-1$ transition tuples, $\{(s_h^\tau, a_h^\tau, s_{h+1}^\tau)\}_{\tau=1}^{t-1}$ and attempt to estimate $\{Q_h^\star\}_{h=1}^H$ via DNNs. Note that, at each time step $h$, these $t-1$ transition tuples are neither independent nor identically distributed due to the interaction with value functions and stochastic transition. To pursue the independence among the transition tuples that is required in our analysis, we follow the *experience replay* scheme [36] that is successfully applied in DQN [4]. The intuition behind experience replay is to break (or weaken) the temporal dependency among the observations for neural networks training. When the replay memory is large (e.g., $10^6$ in DQN [4]), experience replay is close to sampling independent transitions. To be specific, at $t$-th episode, we store transition $\{(s_h^t, a_h^t, r_h, s_{h+1}^t)\}_{h=1}^H$ in the replay memory $\mathcal{D}$, and then sample a mini-batch of *independent* observations from $\mathcal{D}$ with $\{(s_h^{\tau_j}, a_h^{\tau_j}, s_{h+1}^{\tau_j})\}_{(j,h) \in [\tilde{t}] \times [H]}$ for DNNs training. Here the number of mini-batch is denoted as $\tilde{t} := \lceil \varrho t \rceil$ with the mini-batch ratio $\varrho \in (0,1)$, and $\{\tau_j\}_{j=1}^{\tilde{t}}$ is the index for the mino-batch of $\tilde{t}$ independent samples. Note that such independence assumption from experience replay is also used in RL theory, e.g., [37, 44] and theoretically demonstrated to be a good de-correlator [57]. In fact, our analysis only requires independence via experience replay, which is still weaker than the standard iid assumption.

**Value iteration via neural networks:** In our algorithm, we apply the classical least squares value iteration via neural networks for value function learning [28]. We solve the following least squares

---

**Algorithm 1** Value Iteration via DNNs under $\epsilon$-greedy exploration with experience replay

---

1: **Input:** Function class $\mathcal{F}$, the number of episodes $T$, the $\epsilon$-greedy parameter $\epsilon \in (0, 1)$, mini-batch ratio $\varrho \in (0, 1)$.
2: Initialize replay memory $\mathcal{D}$.
3: **for** episode $t = 1, \ldots, T$ **do**
4:     Receive the initial state $s_1^t$.
5:     Set $V_{H+1}^t$ as the zero function.
6:     Set the minibatch size $\tilde{t} := \lceil \varrho t \rceil$ for experience replay.
7:     **for** step $h = H, \ldots, 1$ **do**
8:        Obtain $\widehat{Q}_h^t := \operatorname{argmin}_{f \in \mathcal{F}} \sum_{j=1}^{\tilde{t}} \left[ f(s_h^{\tau_j}, a_h^{\tau_j}) - r_h(s_h^{\tau_j}, a_h^{\tau_j}) - V_{h+1}^t(s_{h+1}^{\tau_j}) \right]^2$.
9:        Obtain $Q_h^t := \widehat{Q}_h^t$ and $V_h^t(\cdot) = \max_{a \in \mathcal{A}} Q_h^t(\cdot, a)$.
10:    **end for**
11:    //$\epsilon$-greedy for exploration
12:    Take the policy $\{\tilde{\pi}_h^t\}_{h=1}^H$ to be greedy policy with probability $1 - \epsilon$ or any policy with probability $\epsilon$.
13:    **for** step $h = 1, \ldots, H$ **do**
14:        Take $a_h^t \sim \tilde{\pi}_h^t(\cdot|s_h^t)$ .
15:        Observe the reward $r_h(s_h^t, a_h^t)$ and obtain the next state $s_{h+1}^t$.
16:    **end for**
17:    //experience replay
18:    Store transition $\{(s_h^t, a_h^t, r_h, s_{h+1}^t)\}_{h=1}^H$ in $\mathcal{D}$.
19:    Sample random mini-batch of transitions from $\mathcal{D}$ with $\tilde{t}$ pairs $\{(s_h^{\tau_j}, a_h^{\tau_j}, s_{h+1}^{\tau_j})\}_{(j,h) \in [\tilde{t}] \times [H]}$.
20: **end for**

---

regression problem via $\tilde{t}$ independent samples

$$\widehat{Q}_h^t = \operatorname*{argmin}_{f \in \mathcal{F}} \widehat{\mathcal{E}}_h^t(f) := \frac{1}{\tilde{t}} \sum_{j=1}^{\tilde{t}} \left[ f(s_h^{\tau_j}, a_h^{\tau_j}) - r_h(s_h^{\tau_j}, a_h^{\tau_j}) - V_{h+1}^t(s_{h+1}^{\tau_j}) \right]^2 . \tag{5}$$

For ease of simplicity for analyses, we directly assume that the global minima solution of problem (5) can be obtained, that follows [58, 52, 59] in deep learning theory. Nevertheless, our result could be extended to allow small optimization error in each episode that will be discussed in Section 6.

Besides, we also need the expectation version of $\widehat{\mathcal{E}}_h^t$ in problem (5) for our analysis. Formally, we assume each state-action pair in the mini-batch is sampled from a respective (unknown) probability measure, i.e., $(s_h^{\tau_j}, a_h^{\tau_j}) \sim \mu_h^{\tau_j}, \forall j \in [\tilde{t}]$, where $\mu_h^{\tau_j} \in \mathscr{P}(\mathcal{S} \times \mathcal{A})$ is from the collection of all probability distribution on $\mathcal{S} \times \mathcal{A}$. Taking the averaged measure $\bar{\mu}_h^{\tilde{t}} := \frac{1}{\tilde{t}} \sum_{j=1}^{\tilde{t}} \mu_h^{\tau_j}$, the expectation of $\widehat{\mathcal{E}}_h^t$ is defined as

$$\mathcal{E}_h^t(f) = \mathbb{E}_{(s_h, a_h) \sim \bar{\mu}_h^{\tilde{t}}, s_{h+1} \sim \mathbb{P}_h(\cdot|s_h, a_h)} \left[ f(s_h, a_h) - r_h(s_h, a_h) - V_{h+1}^t(s_{h+1}) \right]^2 . \tag{6}$$

Note that, $\widehat{Q}_h^t$ in Eq. (5) is not an unbiased estimator of the squared Bellman error minimizer [60, 61]. Indeed, $\mathcal{E}_h^t$ differs from the squared Bellman error because of an extra variance term caused by the stochastic transition [62]. This biased estimation issue can be avoided (or alleviated) in practice by introducing target networks in DQN [63]. Some variants [64] of DQN can also reduce the biased estimate and performs well without target networks. Nevertheless, in our analysis, we center around the uniform bound $\sup_{f \in \mathcal{F}} |\mathcal{E}_h^t(f) - \widehat{\mathcal{E}}_h^t(f)|$ instead of the Bellman error.

$\epsilon$-**greedy exploration:** In order to work in the online setting, we need to ensure that the learner visits "good" state action pairs in the sense that are almost maximizers of the value function for unseen state, *a.k.a.*, exploration. In RL theory, a classical way is to design an optimistic estimate of the value function via a bonus function $b_h^t$ [12, 65] such that $Q_h^t = \min\{\widehat{Q}_h^t + b_h^t, H\}^+$. Instead, we employ the $\epsilon$-greedy exploration that follows DQN-like algorithms. Using the $\epsilon$-greedy exploration will ensure each state-action pair can be visited with positive probability and favor independence among samples. In our algorithm, we directly set $Q_h^t := \min\{\widehat{Q}_h^t, H\}^+$, and then naturally incorporate the truncation operation in neural networks training, see Eqs. (3) and (4).

Based on the above description, our algorithm centers around deep neural function approximation via value iteration under the $\epsilon$-greedy exploration and experience replay under the online setting. This problem setting matches the spirit of practical DQN, which allows for better understanding deep RL.

## 4 Main results

This section presents our results for value iteration under deep (as well as two-layer) ReLU neural networks via the Besov spaces and Barron spaces, respectively. Our theory is based on the independence assumption via experience replay and achieves sublinear regret under the $\epsilon$-greedy exploration.

### 4.1 Efficient value iteration via DNNs in Besov spaces

In this setting, we consider $\widehat{Q}_h^t = \mathrm{argmin}_{f \in \mathcal{F}_{\mathrm{DNN}}} \widehat{\mathcal{E}}_h^t(f)$ in Eq. (5), where $\mathcal{F}_{\mathrm{DNN}}$ is the function space of deep ReLU neural networks defined in Eq. (4). We make the following assumption on the Besov space $\mathcal{B}$, similar to [7, 12], where the Bellman optimality operator maps any bounded value function to a bounded Besov space ball.

**Assumption 1.** *Let $\widetilde{R}$ be a fixed constant. Define $\mathcal{B}_{\widetilde{R}} = \{f \in \mathcal{B}_{p,q}^\alpha(\mathcal{X}) : \|f\|_{\mathcal{B}} \leqslant \widetilde{R}\}$ in the Besov space and assume that for any $h \in [H]$ and $Q \colon \mathcal{S} \times \mathcal{A} \to [0, H]$, we have $\mathbb{T}_h^\star Q \in \mathcal{B}_{\widetilde{R}}$.*

**Remark:** Due to $Q \in [0, H]$, the radius $\widetilde{R}$ in fact depends on $H$, i.e., $\widetilde{R} \asymp H$.

Based on this assumption, we have the following theorem on the regret bound in the Besov space for deep RL under the $\epsilon$-greedy exploration.

**Theorem 1.** *Under Assumption 1 with the smoothness parameter $\alpha > d(1/p - 1/4)_+$ in the Besov space $\mathcal{B}_{p,q}^\alpha(\mathcal{X})$, considering value function learning (5) via DNNs defined by Eq. (4) in Algorithm 1 under the $\epsilon$-greedy exploration and the mini-batch ratio $\varrho \in (0, 1)$, and taking*

$$\text{the depth } L \asymp \frac{d}{2\alpha + d} \log T, \quad \text{the width } m \asymp \frac{d}{2\alpha + d} T^{\frac{d}{2\alpha + d}} \log T, \tag{7}$$

*then given a MDP-dependent constant $K \in [1, H]$, for any $\delta \in (0, 1)$, the total regret can be upper bounded with probability at least $1 - \delta$*

$$Regret(T) \lesssim \left(\frac{\epsilon}{A}\right)^{-\frac{K}{2}} \frac{1}{\sqrt{\varrho}} \left(H^{\frac{3}{2}} T^{\frac{\alpha+d}{2\alpha+d}} \log^3 T + H^2 \sqrt{T} \sqrt{\log\left(\frac{2}{\delta}\right)} \log T\right) + \epsilon H T + \sqrt{T H^3 \log\left(\frac{4}{\delta}\right)}$$

$$\lesssim \widetilde{\mathcal{O}}(H^{\frac{H+4}{H+2}} K^{\frac{2}{K+2}} A^{\frac{K}{K+2}} T^{\frac{\alpha K + (\alpha+d)(K+2)}{(2\alpha+d)(K+2)}}), \quad taking \ \epsilon = \mathcal{O}((HK)^{\frac{2}{K+2}} A^{\frac{K}{K+2}} T^{-\frac{2\alpha}{(2\alpha+d)(K+2)}}). \tag{8}$$

**Remark:** We make the following remarks.
*i)* The constant $K$ describes the "myopic" level of MDPs under the $\epsilon$-greedy policy, e.g., the worst case ($K := H$) under the sparse rewards setting; the benign case $K := c$ (for some small constant $c$) under the helpful dense rewards setting as discussed in [49]. The exponential dependence on $H$ (in the worst case for any MDP) can be avoided at an additional cost of worsening $T$ dependence. In fact, whether in the benign/worst case, the sublinear regret is always achieved under some certain $\epsilon$ values in Eq. (8), which theoretically demonstrates the efficiency of deep RL. Note that the chosen $\epsilon \in (0, 1)$ is always satisfied under a large episode $T$.
*ii)* Clearly, the regret bound is a non-increasing function of the smoothness parameter $\alpha$, which shows that an easier task (i.e., the target Q function is more smooth) leads to regret bounds with faster rates. Specially, if we take $\alpha \to \infty$ (i.e., the target Q function is sufficiently smooth), which holds for linear function approximation

$$\text{Regret}(T) \lesssim \widetilde{\mathcal{O}}(H^{\frac{H+4}{H+2}} K^{\frac{2}{K+2}} A^{\frac{K}{K+2}} T^{\frac{K+1}{K+2}}),$$

which recovers the regret bound $\widetilde{\mathcal{O}}(T^{\frac{K+1}{K+2}})$ in [49, Theorem 3] via Eluder dimension. In the best case ($K = 1$), our regret bound implies $\widetilde{\mathcal{O}}(H^{\frac{4}{3}} A^{\frac{1}{3}} T^{\frac{2}{3}})$ with $H \geqslant 4$, which matches the optimal regret bound for the contextual bandits problem in terms of dependence on $T$ or $A$ under the $\epsilon$-greedy exploration [66]. In the worst case ($K := H$), we can still obtain the sublinear regret at a certain $\widetilde{\mathcal{O}}(T^{\frac{H+1}{H+2}})$ rate.

Theorem 1 demonstrates that the sublinear regret can be achieved by choosing $\mathcal{O}(\log T)$ depth and $\widetilde{\mathcal{O}}(T^{\frac{d}{2\alpha+d}})$ width, but the sublinear regret bound $\widetilde{\mathcal{O}}(T^{\frac{\alpha K+(\alpha+d)(K+2)}{(2\alpha+d)(K+2)}})$ heavily depends on the feature dimension $d$, failing in the *curse of dimensionality*, which appears ineffective on high dimensional data in deep RL. In the next, we consider the Barron spaces, i.e., the "largest" function space for two-layer neural networks to avoid the curse of dimensionality. In this case, the rate of the sublinear regret can get rid of $d$, which is useful for high dimensional data in practical RL.

## 4.2 Efficient value iteration via two-layer neural networks in Barron spaces

As mentioned before, Barron spaces are the "largest" function space for two-layer neural networks. In this setting, we consider $\widehat{Q}_h^t = \mathrm{argmin}_{f \in \mathcal{F}_{\mathrm{SNN}}} \widehat{\mathcal{E}}_h^t(f)$ in Eq. (5), where $\mathcal{F}_{\mathrm{SNN}}$ is the function space of two-layer ReLU neural networks defined in Eq. (3). We give a similar assumption on the Bellman optimality operator in the Barron space.

**Assumption 2.** *Let $\widetilde{R} > 0$ be a fixed constant. Define $\mathcal{P}_{\widetilde{R}} = \{f \in \mathcal{P} : \|f\|_{\mathcal{P}} \leq \widetilde{R}\}$ in the Barron space, and assume that for any $h \in [H]$ and $Q \colon \mathcal{S} \times \mathcal{A} \to [0, H]$, we have $\mathbb{T}_h^\star Q \in \mathcal{P}_{\widetilde{R}}$.*

Based on this assumption, we have the following regret bounds for two-layer ReLU neural networks.

**Theorem 2.** *Under Assumption 2, considering value function learning* (5) *by two-layer ReLU neural networks with width $m$ and bounded $\ell_1$ norm $B$ defined by Eq.* (3) *in Algorithm 1 under the $\epsilon$-greedy exploration and the mini-batch ratio $\varrho \in (0, 1)$, then given a MDP-dependent constant $K \in [1, H]$, for any $\delta \in (0, 1)$, the total regret can be upper bounded with probability at least $1 - \delta$*

$$Regret(T) \lesssim \left(\frac{\epsilon}{A}\right)^{-\frac{K}{2}} \left(\frac{H^2 T^{\frac{3}{4}}}{\sqrt{\varrho}} \left[B(\log d)^{\frac{1}{4}} + \log^{\frac{1}{4}}\left(\frac{4}{\delta}\right)\right] + \frac{H^2 T}{\sqrt{m}}\right) + \epsilon H T + \sqrt{T H^3 \log\left(\frac{4}{\delta}\right)}$$

$$\lesssim \widetilde{\mathcal{O}}(H^{\frac{K+4}{K+2}} T^{\frac{2K+3}{2K+4}}), \quad \text{by taking } m = \Omega(\sqrt{T}) \text{ and } \epsilon = \mathcal{O}\left(H^{\frac{2}{K+2}} T^{-\frac{1}{2(K+2)}}\right).$$

**Remark:** In our result, taking $m = \Omega(\sqrt{T})$ is suffice to achieve the sublinear regret bound $\widetilde{\mathcal{O}}(T^{\frac{2K+3}{2K+4}})$, which also gets rid of the feature dimension $d$, allowing for high-dimensional image data in practice.

# 5 Discussion on architecture guidelines in deep RL

In this section, we present a detailed discussion on how our results provide the architecture guidelines in practical deep RL, in the perspective of the width, the depth, and problem-dependent smoothness of the Q function.

**Width-depth and DQN:** According to Theorem 1, the $\mathcal{O}(\log T)$ depth and $\widetilde{O}(T^{\frac{d}{2\alpha+d}})$ width are enough for sublinear regret in deep RL. Interestingly, we notice that this result is closely matching practical implementation of DQN. For example, the choices of [4] $m = 512$ and $L = 5$ can be explained by our theory, indeed $\log(512) \approx 6$. Specially, when taking $\alpha \to \infty$, this setting holds for linear function approximation. For two-layer neural networks endowed by the Barron space, the curse of dimensionality in terms of width and regret bound can be avoided in Theorem 2, supporting the premise of practical, high-dimensional RL.

**Problem-dependent smoothness and exploration:** The problem-dependent smoothness, determined by $\alpha$, largely affects our regret bounds. The difficulty of a task in deep RL can be defined in two views: one is the smoothness of the target Q function; and the other is the degree of exploration. Intuitively speaking, if a RL task is difficult, then the target Q function is often complicated, and thus admits a relative lower smoothness; or we need conduct more exploration in a complex scenario. Our results coincide with these two views. One hand, the regret bound in Theorem 1 is a non-increasing function of the smoothness parameter $\alpha$. A more difficult task in deep RL (i.e., a smaller $\alpha$) leads to a slower rate of the sublinear regret, which indicates that more episodes are required. On the other hand, Theorem 1 shows that the parameter $\epsilon$ is also a non-increasing function of $\alpha$. That means, a more difficult task in deep RL requires a larger $\epsilon$, i.e., we need conduct exploration more frequently.

Besides, the exploration parameter $\epsilon$ is also affected by $K$ for MDPs with different situations. For example, compared to the best case $K = 1$, more frequent exploration (a larger $\alpha$) is required in MDPs under difficult cases, which coincides with our certain $\epsilon$ value in Theorems 1 and 2.

$$\text{Regret decomp. Lem. 1} \begin{cases} \text{statistical error: Lem. 2 with } \mathcal{O}(\sqrt{H^3 T}) \\ \texttt{Term (i)} \Leftarrow \text{Lem. 5} \begin{cases} \text{generalization} \\ \text{approximation} \end{cases} \Leftarrow \text{Lem. 4: } \|\Gamma_h^t\|^2_{L^2(\mathrm{d}\bar{\mu}_h^{\tilde{t}})} \Leftarrow \text{Lem. 3: } \bar{\mu}_h^{\tilde{t}}(\mathcal{C}) > 0 \\ \epsilon\text{-greedy exploration: } \epsilon H T \end{cases}$$

Figure 1: Proof framework of regret decomposition and transformation of the TD error.

**Width and depth trade-off:** Under a limit parameter budget, according to the width-depth ratio $m/L = T^{\frac{d}{2\alpha+d}}$ in Theorem 1, our theory indicates that less problem-dependent smoothness of Q-function requires DNNs to be wider. In practice, if we work in the limited budget of parameters $N$ in neural networks, e.g., $N \asymp m^2 L$, our theory implies that there is a tradeoff between the depth and width on smoothness, i.e., the depth $L := N^{1/3} T^{-\frac{2d}{3(2\alpha+d)}}$ increasing with $\alpha$ (or $T$) and the width $m = N^{1/3} T^{\frac{d}{3(2\alpha+d)}}$ decreasing with $\alpha$ (or $T$).

Besides, according to the width-depth ratio, it can be found that, the change of $\alpha$ leads to less changes on the depth but more changes on the width. This shows that width and depth admit different levels of parameter sensitivity under the change of problem-dependent smoothness.

## 6 Proof outline

In this section, we outline the proof of our theoretical results presented in Section 4. As mentioned before, the technical challenge in our analysis is how to estimate the TD error without bonus function design. Apart from the regret decomposition, our proof framework includes two main parts: transformation of TD error estimation to generalization bounds, see Figure 1; and generalization bounds on non-iid data in certain Besov/Barron spaces for TD error analysis, see Figure 2. The complete proof is reported in the appendix.

**Regret decomposition:** This part is standard and commonly studied in RL theory, e.g., [65, 7, 12]. We briefly include here for self-completeness. Define the temporal-difference (TD) error as

$$\Gamma_h^t(s, a) = r_h(s, a) + (\mathbb{P}_h V_{h+1}^t)(s, a) - Q_h^t(s, a) = (\mathbb{T}_h V_{h+1}^t)(s, a) - Q_h^t(s, a), \quad \forall (s, a) \in \mathcal{S} \times \mathcal{A}, \tag{9}$$

where $\Gamma_h^t$ is a function on $\mathcal{S} \times \mathcal{A}$ for all $h \in [H]$ and $t \in [T]$. Accordingly, the regret can be decomposed into (*c.f.* Lemma 1)

$$\text{Regret}(T) \leqslant \underbrace{\sum_{t=1}^{T} \sum_{h=1}^{H} \left( \mathbb{E}_{\pi^*}[\Gamma_h^t(s_h, a_h) \,|\, s_1 = s_1^t] - \Gamma_h^t(s_h^t, a_h^t) \right)}_{\texttt{Term (i)}} + \texttt{Term (ii)} + \epsilon H T, \tag{10}$$

where the first term relates to the TD error and the second term is the statistical error based on the standard martingale difference sequences, which can be upper bounded by the Hoeffding-Azuma inequality with $\mathcal{O}(\sqrt{H^3 T})$ regret (*c.f.* Lemma 2). The last term $\epsilon H T$ is due to the $\epsilon$-greedy exploration.

**Transforming TD error to generalization bounds:** To bound the TD error, we first introduce Lemma 3 with $\bar{\mu}_h^{\tilde{t}}(\mathcal{C}) > 0$, where the event $\mathcal{C}$ denotes that all state-action pairs have been visited at all time steps under the $\epsilon$-greedy policy. Then we are able to build the connection between `Term (i)` and $\|\Gamma_h^t\|_{L^2(\mathrm{d}\bar{\mu}_h^{\tilde{t}})}$ in the $L^2(\mathrm{d}\bar{\mu}_h^{\tilde{t}})$-integrable space (*c.f.* Lemma 4). After analysis of $\mathcal{E}_h^t(f)$ in Proposition 1, we transform the estimation of $\|\Gamma_h^t\|_{L^2(\mathrm{d}\bar{\mu}_h^{\tilde{t}})}$ to the following two terms: generalization error and approximation error, respectively (*c.f.* Lemma 5)

$$\|\Gamma_h^t\|^2_{L^2(\mathrm{d}\bar{\mu}_h^{\tilde{t}})} \leqslant \left[ \mathcal{E}_h^t(\widehat{Q}_h^t) - \min_{f \in \mathcal{F}} \mathcal{E}_h^t(f) \right] + \inf_{f \in \mathcal{F}} \|f - \mathbb{T}_h^\star Q_{h+1}^t\|^2_{L^2(\mathrm{d}\bar{\mu}_h^{\tilde{t}})}. \tag{11}$$

where the first term is the generalization error which we elucidate in the next and the second term is the approximation error and can be considered in an $L^p(\mathcal{X})$ space for Besov spaces in Corollary 1. For example, the approximation error in the Besov space admits the certain $\mathcal{O}(N^{-2\alpha/d})$ rate in [30] for deep ReLU networks with $L \asymp \log N$, $S \asymp N$, $m \asymp N \log N$.

$$\begin{cases} \text{Rademacher complexity on non-iid data: Lem. 7} \Leftarrow \text{Lem. 6} \\ \text{two-layer NNs: Thm. 2} \Leftarrow \mathcal{O}(H^2 B \sqrt{\log d/n}) \Leftarrow \text{Lem. 13: Rademacher complexity of Barron spaces} \\ \text{DNNs: Thm. 1} \Leftarrow \mathcal{O}(n^{-\frac{2\alpha}{2\alpha+d}}) \Leftarrow \text{Prop. 2} \Leftarrow \text{Lem. 12 on LRC for Besov spaces} \end{cases}$$

Figure 2: Proof framework of the TD error via generalization bounds on $n$ non-iid data. We denote LRC by local Rademacher complexity for short.

**Generalization bounds on non-iid data:** The key part left is to bound the generalization error on non-i.i.d data for the TD error estimation, see the proof framework in Figure 2. In our proof, we firstly verify that the maximum error in estimating the mean of any function $f \in \mathcal{F}$ can be still bounded by the Rademacher complexity of $\mathcal{F}$ in Lemma 6, and then generalization bounds by Rademacher complexity still holds by Lemma 7 via the averaged measure $\bar{\mu}_h^{\tilde{t}}$, which only requires the data to be independent. These results can be easily extended to local Rademacher complexity.

For deep neural networks, by computing the local Rademacher complexity of $\mathcal{F}_{\text{DNN}}$ in Lemma 12 and choosing proper neural network parameters in Eq. (4), we derive the convergence rate of generalization bounds at a certain $\mathcal{O}(n^{-\frac{2\alpha}{2\alpha+d}})$ rate in Besov spaces (*c.f.* Proposition 2) with $n$ non-iid data. Combining the result of approximation error and taking the depth and width in Eq. (7), Term (i) can be upper bounded with high probability. Finally we conclude the proof of Theorem 1 by combining with the statistical error.

For two-layer neural networks, by computing the Rademacher complexity of $\mathcal{F}_{\text{SNN}}$ in Lemma 13, we obtain the generalization error at a certain $\mathcal{O}(H^2 B \sqrt{\log d/n})$ convergence rate. Combining the result of approximation error in Barron spaces with other terms in the regret decomposition, we conclude the proof of Theorem 2.

**Regret bounds effected by optimization error:** Here we briefly discuss the regret bound affected by a solution (denoted as $\widetilde{Q}_h^t$) that is not a global minimum of problem (5). Assume that the optimization error is small in the functional view, i.e., $\|\widetilde{Q}_h^t - \widehat{Q}_h^t\|_{L^2(\mathrm{d}\bar{\mu}_h^{\tilde{t}})} \leqslant \varepsilon_{\text{opt}}$, that will appear in Eq. (11), and accordingly Term (i) incurs in an extra regret bound $\mathcal{O}(H^2 \log T)$ if we take $\varepsilon_{\text{opt}} := H/\sqrt{\tilde{t}}$. This condition is fair and reasonable as the optimization error decreases with the mini-batch size $\tilde{t}$ for neural network training but requires a refined analysis under non-iid data [67, 68].

## 7 Conclusion

This paper provides an in-depth understanding on neural network function approximation with the $\epsilon$-greedy exploration under the online setting beyond the "linear" regime. Our results provide theoretical guarantees of sublinear regret bounds, and shed light on some guidelines for understanding deep RL in the perspective of the width-depth configuration and the problem-dependent smoothness of RL tasks.

The analysis of this work is built on the $\epsilon$-greedy policy for exploration, which are satisfied in practical cases when employing DQN. Nevertheless, designing a provably efficient exploration mechanism for deep RL could be an interesting future direction in both practice and theory. Besides, our theory requires state-action pairs to be independent, which (approximately) holds via experience replay and could be improved by reverse experience replay [69]. Furthermore, our work is built on the value iteration based algorithm, which is different from practical DQN that adapts Q-learning via one-step gradient descent. Towards a better understanding DQN in terms of Q learning and target networks [43, 44] would be an interesting direction.

## Acknowledgement

The authors would like to thank anonymous reviewers for their constructive suggestions to improve the presentation and point out the independence issue.

This work was supported by SNF project – Deep Optimisation of the Swiss National Science Foundation (SNSF) under grant number 200021_205011; the Enterprise for Society Center (E4S); the European Research Council (ERC) under the European Union's Horizon 2020 research and innovation programme (grant agreement n°725594 - time-data).

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
