## A.1 Besov spaces

Here we briefly introduce a general function space for deep ReLU neural networks according to the "smoothness" of the function, i.e., Besov spaces.

To define Besov functions, we need introduce the modulus of smoothness.

**Definition 1.** *[30, modulus of smoothness] For a function $f \in L^p(\mathcal{X})$ with some $p \in (0, \infty]$, the $k$-th modulus of smoothness of $f$ is defined by*

$$w_{k,p}(f, t) = \sup_{\boldsymbol{h} \in \mathbb{R}^d: \, \|\boldsymbol{h}\|_2 \leqslant t} \|\Delta_{\boldsymbol{h}}^k(f)\|_p \,,$$

*with*

$$\Delta_{\boldsymbol{h}}^k(f)(\boldsymbol{x}) = \begin{cases} \sum_{j=0}^k \binom{k}{j}(-1)^{k-j} f(\boldsymbol{x} + j\boldsymbol{h}) & \text{if } \boldsymbol{x} \in \mathcal{X}, \, \boldsymbol{x} + k\boldsymbol{h} \in \mathcal{X} \,, \\ 0 & \text{otherwise.} \end{cases}$$

The quantity $\Delta_{\boldsymbol{h}}^k(f)$ captures the local oscillation of function $f$ that is not necessarily differentiable. Based on this, the Besov space is defined as below.

**Definition 2.** *[54, 30, Besov space $\mathcal{B}_{p,q}^\alpha(\mathcal{X})$] For $0 < p, q \leqslant \infty$, the smoothness parameter $\alpha > 0$, $k := \lfloor \alpha \rfloor + 1$, define the semi-norm $|\cdot|_{\mathcal{B}_{p,q}^\alpha}$ as*

$$|f|_{\mathcal{B}_{p,q}^\alpha} := \begin{cases} \left( \int_0^\infty (t^{-\alpha} w_{k,p}(f, t))^q \frac{\mathrm{d}t}{t} \right)^{\frac{1}{q}} & (q < \infty) \,, \\ \sup_{t>0} t^{-\alpha} w_{k,p}(f, t) & (q = \infty) \,. \end{cases}$$

*The norm of the Besov space $\mathcal{B}_{p,q}^\alpha(\mathcal{X})$ is defined by $\|f\|_{\mathcal{B}_{p,q}^\alpha} := \|f\|_{L^p(\mathcal{X})} + |f|_{\mathcal{B}_{p,q}^\alpha}$, and the Besov space is $\mathcal{B}_{p,q}^\alpha(\mathcal{X}) = \{ f \in L^p(\mathcal{X}) \mid \|f\|_{\mathcal{B}_{p,q}^\alpha} < \infty \}$.*

The smoothness parameter $\alpha$ indicates which function at a certain smoothness degree can be represented. For example, if $\alpha > d/p$, then the related Besov space is continuously embedded in the set of the continuous functions. However, if $\alpha < d/p$, then the functions in the Besov space are no longer continuous. In particular, the Besov space reduces to the Hölder space $\mathtt{C}^\alpha$ when $p = q = \infty$ and $\alpha$ is a positive non-integer; degenerates to the Sobolev space $\mathtt{W}_2^\alpha$ when $p = q = 2$ and $\alpha$ is a positive integer. The Besov space is more general than these two spaces as it allows for spatially inhomogeneous smoothness with spikes and jumps. More properties of Besov spaces and relations to other function spaces refer to [30] for details.

## A.2 Barron spaces

The study for deep ReLU neural networks is endowed by Besov spaces, but the complete of function space for deep ReLU neural networks to avoid the *curse of dimensionality* is still open. Luckily, the

complete of function space for two-layer neural networks can be conducted by Barron spaces. Here we briefly introduce the basic definition and property of Barron spaces [55, 31].

We consider a typical two-layer neural network $f(\boldsymbol{x}) = \frac{1}{m}\sum_{k=1}^{m} b_k \sigma(\boldsymbol{w}_k^\top \boldsymbol{x} + c_k)$, where $m$ is the number of neurons in the hidden layer and $\sigma(x) = \max\{x, 0\}$ is the ReLU activation function used in this work. Accordingly, the two-layer neural network admits the following representation

$$f(\boldsymbol{x}) = \int_\Omega b\sigma\left(\boldsymbol{w}^\top \boldsymbol{x} + c\right)\rho(\mathrm{d}b, \mathrm{d}\boldsymbol{w}, \mathrm{d}c), \quad \boldsymbol{x} \in \mathcal{X}, \tag{12}$$

where $\Omega = \mathbb{R} \times \mathbb{R}^d \times \mathbb{R}$ and $\rho$ is a probability measure over $\Omega$. Then the Barron space [31] endowed by the $p$-Barron norm with $p \in [1, +\infty]$ is defined as

$$\widetilde{\mathcal{P}}_p = \left\{ f \text{ admits Eq. (12)} : \|f\|_{\widetilde{\mathcal{P}}_p} = \inf_\rho \left\{ \mathbb{E}_\rho |b|^p (\|\boldsymbol{w}\|_1 + |c|)^p \right\}^{1/p} < \infty \right\}.$$

Specifically, when using ReLU, these function spaces under different $p$ are the same, i.e., $\widetilde{\mathcal{P}}_1 = \widetilde{\mathcal{P}}_2 = \cdots = \widetilde{\mathcal{P}}_\infty$, and thus we directly use $\widetilde{\mathcal{P}}$ for short. The is the main reason why we study ReLU activation functions in this work. Besides, the Barron norm is close to the $\ell_1$-path norm [70]

$$\|f\|_{\widetilde{\mathcal{P}}} \leqslant \|f\|_{\mathcal{P}} := \frac{1}{m}\sum_{k=1}^{m} |b_k|(\|\boldsymbol{w}_k\|_1 + |c_k|) \leqslant 2\|f\|_{\widetilde{\mathcal{P}}}.$$

Based on this, for description simplicity, we do not strictly distinguish the Barron norm and the $\ell_1$-path norm, and regard $\|f\|_{\mathcal{P}}$ as the discrete version of the Barron norm.

As suggested by [55], Barron space can be regarded as the *largest* function space for two layer neural networks in two folds [31]: 1) *direct approximation:* Any function in Barron spaces can be efficiently approximated by two-layer neural networks with bounded $\ell_1$ path norm at $\mathcal{O}(1/m)$ rate without *curse of dimensionality*; 2) *inverse approximation:* Any continuous function that can be efficiently approximated by two-layer neural networks with bounded $\ell_1$-path norm belongs to a Barron space.

# B  Regret decomposition

We present the regret decomposition under the $\epsilon$-greedy policy by constructing the martingale difference sequence and giving error bounds for this. Apart from an extra $\epsilon H T$ regret, this decomposition result appears in [65, 12], and we include them here just for self-completeness.

To establish the regret decomposition, we need some notations. Remember the definition of the regret, $\tilde{\pi}^t$ is the $\epsilon$-greedy policy and $\pi^t$ is the greedy policy at the $t$-th episode, and then we have

$$\text{Regret}(T) = \sum_{t=1}^{T}\left[V_1^\star(s_1^t) - V_1^{\pi^t}(s_1^t)\right] + \sum_{t=1}^{T}\left[V_1^{\pi^t}(s_1^t) - V_1^{\tilde{\pi}^t}(s_1^t)\right]$$

$$\leqslant \sum_{t=1}^{T}\left[V_1^\star(s_1^t) - V_1^{\pi^t}(s_1^t)\right] + \epsilon H T,$$

where $\epsilon H T$ stems from the fact that the return of greedy and $\epsilon$-greedy policies can differ at most $\epsilon H$ in each episode. In the next, we aim to estimate the first term in the above equation. It involves the greedy policy $\pi^t$ at the $t$-th episode, which leads to a trajectory $\{(s_h^t, a_h^t)\}_{h=1}^{H}$. Note that this trajectory is different from Algorithm 1 that uses the $\epsilon$-greedy policy but we use the same notation on state-action pairs for notational simplicity in this section.

Following [65, 12], we define two quantities $\zeta_{t,h}^1, \zeta_{t,h}^2 \in \mathbb{R}$ for any $h \in [H]$ and $t \in [T]$ based on the greedy policy

$$\begin{aligned}\zeta_{t,h}^1 &:= \left[V_h^t(s_h^t) - V_h^{\pi^t}(s_h^t)\right] - \left[Q_h^t(s_h^t, a_h^t) - Q_h^{\pi^t}(s_h^t, a_h^t)\right], \\ \zeta_{t,h}^2 &:= \left[(\mathbb{P}_h V_{h+1}^t)(s_h^t, a_h^t) - (\mathbb{P}_h V_{h+1}^{\pi^t})(s_h^t, a_h^t)\right] - \left[V_{h+1}^t(s_{h+1}^t) - V_{h+1}^{\pi^t}(s_{h+1}^t)\right].\end{aligned} \tag{13}$$

By definition, $\zeta_{t,h}^1$ depends on the randomness of choosing an action $a_h^t \sim \pi_h^t(\cdot|s_h^t)$; and $\zeta_{t,h}^2$ captures the stochastic transition, i.e., the randomness of drawing the next state $s_{h+1}^t$ from $\mathbb{P}_h(\cdot|s_h^t, a_h^t)$. Based on the following definition of filtration, $\{\zeta_{t,h}^1, \zeta_{t,h}^2\}$ forms a bounded martingale difference sequence.

**Definition 3.** *[65, Filtration] For any $(t, h) \in [T] \times [H]$, define $\sigma$-algebras $\mathcal{M}_{t,h,1}$ and $\mathcal{M}_{t,h,2}$ generated by the following respective state-action sequence as*

$$\mathcal{M}_{t,h,1} := \sigma\big(\{(s_i^\tau, a_i^\tau)\}_{(\tau,i) \in [t-1] \times [H]} \cup \{(s_i^t, a_i^t)\}_{i \in [h]}\big),$$
$$\mathcal{M}_{t,h,2} := \sigma\big(\{(s_i^\tau, a_i^\tau)\}_{(\tau,i) \in [t-1] \times [H]} \cup \{(s_i^t, a_i^t)\}_{i \in [h]} \cup \{s_{h+1}^t\}\big), \tag{14}$$

*where we identify $\mathcal{F}_{t,0,2}$ with $\mathcal{M}_{t-1,H,2}$ for all $t \geqslant 2$ and let $\mathcal{M}_{1,0,2}$ be the empty set. Further, for any $t \in [T]$, $h \in [H]$ and $m \in [2]$, we define the time-step index $\tau(t, h, m)$ as*

$$\tau(t, h, m) = (t - 1) \cdot 2H + (h - 1) \cdot 2 + m, \tag{15}$$

*which offers an partial ordering over the triplets $(t, h, m) \in [T] \times [H] \times [2]$. Moreover, according to Eq. (14), for any $(t, h, m)$ and $(t', h', m')$ satisfying $\tau(k, h, m) \leqslant \tau(k', h', m')$, it holds that $\mathcal{M}_{k,h,m} \subseteq \mathcal{M}_{k',h',m'}$. Thus, the sequence of $\sigma$-algebras $\{\mathcal{M}_{t,h,m}\}_{(t,h,m) \in [T] \times [H] \times [2]}$ forms a filtration.*

Accordingly, we have the following regret decomposition result.

**Lemma 1** (Regret Decomposition [65, 12]). *Recall the definition of the temporal-difference error $\Gamma_h^t : \mathcal{S} \times \mathcal{A} \to$ in Eq. (9) for all $(t, h) \in [T] \times [H]$, then the regret can be decomposed as*

$$Regret(T) \leqslant \underbrace{\sum_{t=1}^{T} \sum_{h=1}^{H} \big[\mathbb{E}_{\pi^\star}[\Gamma_h^t(s_h, a_h) \,|\, s_1 = s_1^t] - \Gamma_h^t(s_h^t, a_h^t)\big]}_{\texttt{Term (i)}} + \underbrace{\sum_{t=1}^{T} \sum_{h=1}^{H} (\zeta_{t,h}^1 + \zeta_{t,h}^2)}_{\texttt{Term (ii)}}$$

$$+ \underbrace{\sum_{t=1}^{T} \sum_{h=1}^{H} \mathbb{E}_{\pi^\star}\big[\langle Q_h^t(s_h, \cdot), \pi_h^\star(\cdot \,|\, s_h) - \pi_h^t(\cdot | s_h)\rangle_{\mathcal{A}} \big| s_1 = s_1^t\big]}_{\texttt{Term (iii)} \leqslant 0} + \epsilon H T, \tag{16}$$

*where $\zeta_{t,h}^1$ and $\zeta_{t,h}^2$ are defined in Eq. (13).*

*Proof.* Remember the definition of the regret, $\tilde{\pi}^t$ is the $\epsilon$-greedy policy and $\pi^t$ is the greedy policy at the $t$-th episode, and then we have

$$\text{Regret}(T) = \sum_{t=1}^{T} \big[V_1^\star(s_1^t) - V_1^{\pi^t}(s_1^t)\big] + \sum_{t=1}^{T} \big[V_1^{\pi^t}(s_1^t) - V_1^{\tilde{\pi}^t}(s_1^t)\big]$$

$$\leqslant \sum_{t=1}^{T} \underbrace{V_1^\star(s_1^t) - V_1^t(s_1^t)}_{(*)} + \sum_{t=1}^{T} \underbrace{V_1^t(s_1^t) - V_1^{\pi^t}(s_1^t)}_{(**)} + \epsilon H T, \tag{17}$$

where the first term (*) can be bounded by [65, 12]

$$V_1^\star(s_1^t) - V_1^t(s_1^t) = \sum_{h=1}^{H} \big[\mathbb{E}_{\pi^\star}[\Gamma_h^t(s_h, a_h) \,|\, s_1 = s_1^t]$$

$$+ \underbrace{\sum_{h=1}^{H} \mathbb{E}_{\pi^\star}\big[\langle Q_h^t(s_h, \cdot), \pi_h^\star(\cdot \,|\, s_h) - \pi_h^t(\cdot | s_h)\rangle_{\mathcal{A}} \big| s_1 = s_1^t\big]}_{\leqslant 0}, \quad \forall t \in [T],$$

where we use the fact that $\pi^t$ is the greedy policy with respect to $Q_h^t$ for any $(t, h) \in [T] \times [H]$. The second term (**) is also bounded by [65, 12]

$$V_1^t(s_1^t) - V_1^{\pi^t}(s_1^t) = \sum_{h=1}^{H} (\zeta_{t,h}^1 + \zeta_{t,h}^2) - \sum_{h=1}^{H} \Gamma_h^t(s_h^t, a_h^t), \quad \forall t \in [T].$$

Finally, we conclude the proof. $\qquad\square$

In the next, it is natural to employ Azuma-Hoeffding inequality for martingale difference sequences as below.

**Lemma 2.** *[65, statistical error] For $\zeta_{t,h}^1$ and $\zeta_{t,h}^2$ defined in Eq. (13) and for any $\delta \in (0,1)$, with probability at least $1 - \delta$, we have*

$$\sum_{t=1}^{T}\sum_{h=1}^{H}(\zeta_{t,h}^1 + \zeta_{t,h}^2) \lesssim \sqrt{TH^3 \log(2/\delta)}\,.$$

## C   Proofs of transformation on the temporal difference error

In this section, we aim to transform the temporal difference error in `Term (i)` to generalization bounds. This is the key part in our proof without bonus function design.

### C.1   TD error under the averaged measure

Here we build the connection between `Term (i)` in the regret decomposition and the TD error $\Gamma_h^t$ in the $L^2(\mathrm{d}\bar{\mu}_h^{\tilde{t}})$-integrable space.

To this end, we need study the relationship between $L^2(\mathrm{d}\mu)$-norm and $L^\infty$-norm, where $\mu$ can be any probability measure over $\mathcal{S} \times \mathcal{A}$. For any $f \in L^2(\mathrm{d}\mu)$ with $\delta \leqslant \|f\|_\infty$, denote

$$\mathcal{G}_\delta := \{(s,a) : |f(s,a)| \geqslant \|f\|_\infty - \delta\}, \quad \forall (s,a) \in \mathcal{S} \times \mathcal{A}, \tag{18}$$

then we have the following lemma that $\bar{\mu}_h^{\tilde{t}}(\mathcal{G}_\delta)$ can be lower bounded under the $\epsilon$-greedy policy.

**Lemma 3.** *Under the $\epsilon$-greedy policy, considering the set in Eq. (18) and the averaged measure $\bar{\mu}_h^{\tilde{t}}$ based on a mini-batch of $\tilde{t}$ historical state-action pairs, we have*

$$\bar{\mu}_h^{\tilde{t}}(\mathcal{G}_\delta) \geqslant \Omega\left(\left(\frac{\epsilon}{A}\right)^H\right), \quad \forall \epsilon \in (0,1) \ \text{and} \ \delta \geqslant 0\,.$$

**Remark:** Clearly, in the best case, we have $\bar{\mu}_h^{\tilde{t}}(\mathcal{G}_\delta) \geqslant \Omega\left(\frac{\epsilon}{A}\right)$. Accordingly, we denote $K \in [1, H]$ as a MDP-dependent constant to describe the "myopic" level of MDPs [49] such that $\bar{\mu}_h^{\tilde{t}}(\mathcal{G}_\delta) \geqslant \Omega\left((\epsilon/A)^K\right)$.

*Proof.* For any $f \in L^2(\mathrm{d}\mu)$ with $\delta \leqslant \|f\|_\infty$, we have

$$\|f\|_{L^2(\mathrm{d}\mu)} \geqslant \left(\int_{\mathcal{G}_\delta}(\|f\|_\infty - \delta)^2 \mathrm{d}\mu\right)^{1/2} = (\|f\|_\infty - \delta)[\mu(\mathcal{G}_\delta)]^{1/2}\,, \tag{19}$$

which is also valid to $\bar{\mu}_h^{\tilde{t}}$. Clearly, $\bar{\mu}_h^{\tilde{t}}(\mathcal{G}_\delta) \in [0,1]$.

To prove $\bar{\mu}_h^{\tilde{t}}(\mathcal{G}_\delta) > 0$ with the lower bound, we consider **the worst case** with $\delta = 0$ and every time step taking non-greedy action with probability $\epsilon$. That means, we need to find the optimal state-action pair in Eq. (18), which can be achieved by the fact that all state-action pairs have been visited at all time steps. It is clear that the cardinality of $\mathcal{G}_\delta$ is a non-decreasing function of $\delta$. Accordingly, there exists $j \in [\tilde{t}]$ such that

$$\bar{\mu}_h^{\tilde{t}}(\mathcal{G}_\delta) \geqslant \bar{\mu}_h^{\tilde{t}}(\mathcal{G}_0) \geqslant \min_{(s,a,h)} \mu_h^{\pi_h^{\tau_j}}(s_h^{\tau_j}, a_h^{\tau_j})\,,$$

where $\mu_h^{\pi_h^{\tau_j}}$ is the occupancy measure of the policy $\pi_h^{\tau_j}$ at the $h$-step and $t$-th episode. Accordingly, $\mu_h^{\pi_h^{\tau_j}}$ admits the following representation

$$\mu_h^{\pi_h^{\tau_j}}\left(s_h^{\tau_j}, a_h^{\tau_j}\right) = \sum_{s_1^{\tau_j},\dots,s_{h-1}^{\tau_j}} \left(\prod_{i=1}^{h-1}\sum_{a\in\mathcal{A}} \Pr\left(\pi_h^\tau\left(s_i^{\tau_j}\right) = a\right) \mathbb{P}_i\left(s_{i+1}^{\tau_j} \mid s_i^{\tau_j}, a\right)\right) \Pr\left(\pi_h^{\tau_j}\left(s_h^{\tau_j}\right) = a_h^{\tau_j}\right)\,.$$

Accordingly, in the worst case, at every time step we take any one action with probability $\epsilon/A$ such that

$$\mu_h^{\pi_h^{\tau_j}}\left(s_h^{\tau_j}, a_h^{\tau_j}\right) \geqslant \Omega\left(\left(\frac{\epsilon}{A}\right)^h\right) \geqslant \Omega\left(\left(\frac{\epsilon}{A}\right)^H\right)\,,$$

which implies that

$$\bar{\mu}_h^{\tilde{t}}(\mathcal{G}_\delta) \geqslant \Omega\left(\left(\frac{\epsilon}{A}\right)^H\right),$$

and accordingly we conclude the proof. $\qquad\square$

**Lemma 4.** *Given a MDP-dependent constant $K \in [1, H]$, for the temporal-difference error $\Gamma_h^t$ defined in Eq. (9) for all $(t, h) \in [T] \times [H]$, under the $\epsilon$-greedy policy, then* Term (i) *can be upper bounded by*

$$\texttt{Term (i)} \lesssim \left(\frac{\epsilon}{A}\right)^{-\frac{K}{2}} \sqrt{T} \sum_{h=1}^{H} \sqrt{\sum_{t=1}^{T} \|\Gamma_h^t\|_{L^2(\mathrm{d}\bar{\mu}_h^{\tilde{t}})}^2} + \mathcal{O}(H\sqrt{T}).$$

*Proof.* According to the definition of Term (i) in Lemma 1, we have

$$\texttt{Term (i)} \leqslant \sum_{t=1}^{T}\sum_{h=1}^{H}\left(\mathbb{E}_{\pi^\star}\left[\left|\Gamma_h^t(s_h, a_h)\right|\Big|s_1 = s_1^t\right] + \left|\Gamma_h^t(s_h^t, a_h^t)\right|\right)$$

$$\leqslant 2\sum_{t=1}^{T}\sum_{h=1}^{H}\|\Gamma_h^t\|_\infty \quad \text{[hold for any } (s, a) \in \mathcal{S} \times \mathcal{A}] \tag{20}$$

$$\leqslant 2\sum_{t=1}^{T}\sum_{h=1}^{H}\left(\frac{\|\Gamma_h^t\|_{L^2(\mathrm{d}\bar{\mu}_h^{\tilde{t}})}}{\sqrt{\bar{\mu}_h^{\tilde{t}}(\mathcal{G}_\delta)}} + \delta\right) \quad \text{[taking } \mu := \bar{\mu}_h^{\tilde{t}} \text{ in Eq. (19)]}.$$

Furthermore, by taking $\delta := t^{-1/2}$ such that $\int_1^T t^{-1/2}\mathrm{d}t = \mathcal{O}(\sqrt{T})$, and using $\bar{\mu}_h^{\tilde{t}}(\mathcal{G}_\delta) \geqslant \Omega\left((\epsilon/A)^K\right)$ with $K \in [1, H]$ in Lemma 3, the above equation can be further expressed as

$$\texttt{Term (i)} \lesssim \left(\frac{\epsilon}{A}\right)^{-\frac{K}{2}} \sum_{t=1}^{T}\sum_{h=1}^{H}\|\Gamma_h^t\|_{L^2(\mathrm{d}\bar{\mu}_h^{\tilde{t}})} + \mathcal{O}(H\sqrt{T}) \quad \text{[using Lemma 3]}$$

$$\leqslant \left(\frac{\epsilon}{A}\right)^{-\frac{K}{2}} \sum_{h=1}^{H}\sqrt{T}\sqrt{\sum_{t=1}^{T}\|\Gamma_h^t\|_{L^2(\mathrm{d}\bar{\mu}_h^{\tilde{t}})}^2} + \mathcal{O}(H\sqrt{T}), \quad \text{[using elementary inequality]}$$

which concludes the proof. $\qquad\square$

## C.2 Connection between the TD error and generalization bounds

Based on Lemma 4, the key issue left is to bound $\sum_{t=1}^{T}\|\Gamma_h^t\|_{L^2(\mathrm{d}\bar{\mu}_h^{\tilde{t}})}^2 \lesssim o(T)$ for a sublinear regret. To this end, we build the connection between $\|\Gamma_h^t\|_{L^2(\mathrm{d}\bar{\mu}_h^{\tilde{t}})}^2$ and generalization bounds. We first the study the decomposition of $\mathcal{E}_h^t(f)$ in Eq. (6) by the following proposition: there exists an extra variance term in the expected risk $\mathcal{E}_h^t(f)$.

**Proposition 1.** *According to the definition of $\mathcal{E}_h^t(f)$ in Eq. (6), then we have*

$$\mathcal{E}_h^t(f) = \underbrace{\|f - \mathbb{T}_h V_{h+1}^t\|_{L^2(\mathrm{d}\bar{\mu}_h^{\tilde{t}})}^2}_{:=\bar{\mathcal{E}}_h^t(f)} + \mathbb{E}_{(s_h, a_h) \sim \bar{\mu}_h^{\tilde{t}}, s_{h+1} \sim \mathbb{P}_h(\cdot|s_h, a_h)}\texttt{Var}[V_{h+1}^t(s_{h+1})], \tag{21}$$

*where the variance* $\texttt{Var}[V_{h+1}^t(s_{h+1})] := \left[\mathbb{E}_{s_{h+1}}[V_{h+1}^t(s_{h+1})] - V_{h+1}^t(s_{h+1})\right]^2$.

*Proof.* Denote $s' := s_{h+1}$ for short, we expand $\mathcal{E}_h^t(f)$ as the following expression

$$\mathcal{E}_h^t(f) = \mathbb{E}_{(s_h, a_h) \sim \bar{\mu}_h^{\tilde{t}}, s'}\left[f(s_h, a_h) - r_h(s_h, a_h) - \mathbb{E}_{s'}V_{h+1}^t(s') + \mathbb{E}_{s'}V_{h+1}^t(s') - V_{h+1}^t(s')\right]^2$$

$$= \mathbb{E}_{(s_h, a_h) \sim \bar{\mu}_h^{\tilde{t}}}\left[f(s_h, a_h) - r_h(s_h, a_h) - \mathbb{E}_{s'}V_{h+1}^t(s')\right]^2 + \mathbb{E}_{(s_h, a_h) \sim \bar{\mu}_h^{\tilde{t}}, s'}\texttt{Var}[V_{h+1}^t(s')]$$

$$= \|f - \mathbb{T}_h V_{h+1}^t\|_{L^2(\mathrm{d}\bar{\mu}_h^{\tilde{t}})}^2 + \mathbb{E}_{(s_h, a_h) \sim \bar{\mu}_h^{\tilde{t}}, s_{h+1} \sim \mathbb{P}_h(\cdot|s_h, a_h)}\texttt{Var}[V_{h+1}^t(s_{h+1})],$$

where we use $\mathbb{E}_{s' \sim \mathbb{P}_h(\cdot|s_h, a_h)}\left[\mathbb{E}_{s'}[V_{h+1}^t(s')] - V_{h+1}^t(s')\right] = 0$ and conclude the proof. $\qquad\square$

According to the decomposition of $\mathcal{E}_h^t(f)$ Proposition 1, $\bar{\mathcal{E}}_h^t(f)$ in Eq. (21) is close to the squared Bellman error [61]. We are able to transform the estimation of the TD error to generalization error and approximation error as below.

**Lemma 5.** *For the temporal-difference error $\Gamma_h^t$ defined in Eq. (9) for all $(t,h) \in [T] \times [H]$, it can be upper bounded in the $L^2(\mathrm{d}\mu_h^{\tilde{t}})$ space with*

$$\|\Gamma_h^t\|_{L^2(\mathrm{d}\bar{\mu}_h^{\tilde{t}})}^2 \leqslant \left[ \mathcal{E}_h^t(\widehat{Q}_h^t) - \min_{f \in \mathcal{F}} \mathcal{E}_h^t(f) \right] + \inf_{f \in \mathcal{F}} \|f - \mathbb{T}_h^\star Q_{h+1}^t\|_{L^2(\mathrm{d}\bar{\mu}_h^{\tilde{t}})}^2,$$

*where the first term is the generalization error of $\widehat{Q}_h^t$, the second term is the approximation error in the function class $\mathcal{F}$.*

*Proof.* According to the definition of the TD error $\Gamma_h^t$ and taking $f := \widehat{Q}_h^t$ in Eq. (21) given by Proposition 1, we have

$$\begin{aligned}
\mathcal{E}_h^t(\widehat{Q}_h^t) &= \|\widehat{Q}_h^t - \mathbb{T}_h V_{h+1}^t\|_{L^2(\mathrm{d}\bar{\mu}_h^{\tilde{t}})}^2 + \mathbb{E}_{(s_h,a_h) \sim \bar{\mu}_h^{\tilde{t}}, s_{h+1} \sim \mathbb{P}_h(\cdot|s_h,a_h)} \mathrm{Var}[V_{h+1}^t(s_{h+1})] \\
&= \frac{1}{\tilde{t}} \sum_{j=1}^{\tilde{t}} \|\widehat{Q}_h^t - \mathbb{T}_h V_{h+1}^t\|_{L^2(\mathrm{d}\mu_h^{\tau_j})}^2 + \mathbb{E}_{(s_h,a_h) \sim \bar{\mu}_h^{\tilde{t}}, s_{h+1} \sim \mathbb{P}_h(\cdot|s_h,a_h)} \mathrm{Var}[V_{h+1}^t(s_{h+1})] \\
&= \frac{1}{\tilde{t}} \sum_{j=1}^{\tilde{t}} \|\Gamma_h^t\|_{L^2(\mathrm{d}\mu_h^{\tau_j})}^2 + \mathbb{E}_{(s_h,a_h) \sim \bar{\mu}_h^{\tilde{t}}, s_{h+1} \sim \mathbb{P}_h(\cdot|s_h,a_h)} \mathrm{Var}[V_{h+1}^t(s_{h+1})],
\end{aligned} \tag{22}$$

where the second equality holds by the definition of the averaged measure $\bar{\mu}_h^{\tilde{t}} = \frac{1}{\tilde{t}} \sum_{j=1}^{\tilde{t}} \mu_h^{\tau_j}$; and we use $Q_h^t = \widehat{Q}_h^t$ in the last equality as the truncation operation has been given in function classes, see Eqs. (3) and (4). Then, taking the infimum on both sides of Eq. (21), we have

$$\begin{aligned}
\min_{f \in \mathcal{F}} \mathcal{E}_h^t(f) &= \inf_{f \in \mathcal{F}} \|f - \mathbb{T}_h V_{h+1}^t\|_{L^2(\mathrm{d}\bar{\mu}_h^{\tilde{t}})}^2 + \mathbb{E}_{(s_h,a_h) \sim \bar{\mu}_h^{\tilde{t}}, s_{h+1} \sim \mathbb{P}_h(\cdot|s_h,a_h)} \mathrm{Var}[V_{h+1}^t(s_{h+1})] \\
&= \inf_{f \in \mathcal{F}} \|f - \mathbb{T}_h^\star Q_{h+1}^t\|_{L^2(\mathrm{d}\bar{\mu}_h^{\tilde{t}})}^2 + \mathbb{E}_{(s_h,a_h) \sim \bar{\mu}_h^{\tilde{t}}, s_{h+1} \sim \mathbb{P}_h(\cdot|s_h,a_h)} \mathrm{Var}[V_{h+1}^t(s_{h+1})],
\end{aligned} \tag{23}$$

where the second equality holds by $V_{h+1}^t(s_{h+1}) = \max_{a \in \mathcal{A}} Q_{h+1}^t(s_{h+1}, a)$.

Combining Eqs. (22) and (23), we have

$$\|\Gamma_h^t\|_{L^2(\mathrm{d}\bar{\mu}_h^{\tilde{t}})}^2 = \mathcal{E}_h^t(\widehat{Q}_h^t) - \min_{f \in \mathcal{F}} \mathcal{E}_h^t(f) + \inf_{f \in \mathcal{F}} \|f - \mathbb{T}_h^\star Q_{h+1}^t\|_{L^2(\mathrm{d}\bar{\mu}_h^{\tilde{t}})}^2, \tag{24}$$

which concludes the proof. $\qquad\square$

Based on Lemma 5, we have the following corollary if we consider the approximation error in $L^p(\mathcal{X})$-integrable space, which is needed for our results on deep ReLU neural networks.

**Corollary 1.** *Under the same setting of Lemma 5, we have*

$$\|\Gamma_h^t\|_{L^2(\mathrm{d}\bar{\mu}_h^{\tilde{t}})}^2 \lesssim \left[ \mathcal{E}_h^t(\widehat{Q}_h^t) - \min_{f \in \mathcal{F}} \mathcal{E}_h^t(f) \right] + \inf_{f \in \mathcal{F}} \|f - \mathbb{T}_h^\star Q_{h+1}^t\|_{L^4(\mathcal{X})}^2.$$

*Proof.* Following the proof of Lemma 5, this result can be easily obtained by Cauchy-Schwartz inequality. To be specific, for any probability measure $\mu$, we have

$$\|f\|_{L^p(\mathrm{d}\mu)} \leqslant \|f\|_{L^{2p}(\mathcal{X})} \left( \int_{\mathcal{X}} |g(\boldsymbol{x})|^2 \mathrm{d}\boldsymbol{x} \right)^{\frac{1}{2p}} \lesssim \|f\|_{L^{2p}(\mathcal{X})},$$

where $g$ is the probability density function associated with the probability measure $\mu$. Note that the result here still holds true for the approximation error in $L^\infty(\mathcal{X})$ if we use Hölder inequality, but this condition is much stronger as it requires the target Q function to be continuous. $\qquad\square$

# D Generalization bounds on non-iid data

In this section, we prove that the traditional Rademacher complexity is still valid for independent but non-identically distributed data under a well-defined measure. Similarly, such result is also valid to local Rademacher complexity. The key fact is that, the classical Rademacher complexity [71] is still valid as McDiarmid's bound only requires the independent property.

For description simplicity, we consider a general setting beyond our reinforcement learning task, i.e., learning with $n$ independent but non-identical distributed data $X = \{\boldsymbol{x}_i\}_{i=1}^n$ in $\mathbb{R}^d$ with $\boldsymbol{x}_i \sim \mu_i, \forall i \in [n]$. Define the average measure $\bar{\mu} := \frac{1}{n}\sum_{i=1}^n \mu_i$, we have

$$\mathbb{E}_{\boldsymbol{x}\sim\bar{\mu}}[f(\boldsymbol{x})] = \frac{1}{n}\sum_{i=1}^n \int_{\mathbb{R}^d} f(\boldsymbol{x})\mathrm{d}\mu_i(\boldsymbol{x}) = \frac{1}{n}\sum_{i=1}^n \mathbb{E}_{\boldsymbol{x}\sim\mu_i}[f(\boldsymbol{x})]. \tag{25}$$

Accordingly, the *empirical Rademacher complexity* of a function class $\mathcal{F}$ on the sample set $X$ is defined as

$$\widehat{\mathcal{R}}_n(\mathcal{F}, X) = \frac{1}{n}\mathbb{E}_{\boldsymbol{\xi}}\left[\sup_{f\in\mathcal{F}}\sum_{i=1}^n \xi_i f(\boldsymbol{x}_i)\right], \tag{26}$$

where the expectation is taken over $\boldsymbol{\xi} = \{\xi_1, \xi_2, \cdots, \xi_n\}$, i.e., Rademacher random variables, with $\Pr(\xi_i = 1) = \Pr(\xi_i = -1) = 1/2$. The related *Rademacher complexity* under our non-iid setting is defined as

$$\mathcal{R}_n(\mathcal{F}) = \mathbb{E}_{\boldsymbol{x}_1\sim\mu_1, \cdots, \boldsymbol{x}_n\sim\mu_n}\left[\frac{1}{n}\mathbb{E}_{\boldsymbol{\xi}}\left[\sup_{f\in\mathcal{F}}\sum_{i=1}^n \xi_i f(\boldsymbol{x}_i)\right]\right],$$

where the expectation is taken over $\{\boldsymbol{x}_i\}_{i=1}^n$ with respect to each probability measure $\{\mu_i\}_{i=1}^n$. This definition follows the classical *Rademacher complexity* [71] on iid samples to intuitively indicates how expressive the function class is. Besides, in our proof, we also need a notation of *local Rademacher complexity* on a set of vectors, where "local" means that the class over which the Rademacher process is defined is a subset of the original class. Following the same style with Rademacher complexity, the local Rademacher complexity under the non-iid setting is defined as $\mathcal{R}_n\{f \in \mathcal{F} : \mathbb{E}_{\bar{\mu}}f^2 \leqslant R\}$, and the empirical local Rademacher complexity is defined as $\widehat{\mathcal{R}}_n\{f \in \mathcal{F} : P_n f^2 \leqslant R\}$, where we denote $P_n f := \frac{1}{n}\sum_{i=1}^n f(\boldsymbol{x}_i)$ for short.

Besides, Rademacher complexity is also related to covering number, a metric for estimation of a hypothesis space. Here we give the definition of covering number, that is also used in this work.

**Definition 4.** *[72, Definition 5.1, covering number] Let $(\mathcal{F}, \|\cdot\|)$ be a norm space. A $\delta$-cover of the set $\mathcal{F}$ with respect to $\|\cdot\|$ is a set $\{\theta_1, \cdots, \theta_n\} \subseteq \mathcal{F}$ such that for each $\theta \in \mathcal{F}$, there exists some $i \in [n]$ such that $\|\theta - \theta_i\| \leqslant \delta$. The $\delta$-covering number $\mathcal{N}(\delta, \mathcal{F}, \|\cdot\|)$ is the cardinality of the minimal $\delta$-cover.*

In this work, we consider the covering number with two types of norms, one is $\mathcal{N}(\epsilon, \mathcal{F}, \|\cdot\|_\infty)$ and the other is $\mathcal{N}(\epsilon, \mathcal{F}, \|\cdot\|_2) := \sup_n \sup_{P_n} \mathcal{N}(\epsilon, \mathcal{F}, \|\cdot\|_{L_2(P_n)})$ [73].

## D.1 Rademacher complexity on non-iid data

Based on the definition of Rademacher complexity and its empirical version, we have the following lemma.

**Lemma 6.** *Let $X = \{\boldsymbol{x}_i\}_{i=1}^n$ be an independent but non-identical distributed data set with $\boldsymbol{x}_i \sim \mu_i, \forall i \in [n]$, and $R_n(\mathcal{F})$ be the Rademacher complexity of the function class $\mathcal{F}$ on $X$, denote the averaged probability measure as $\bar{\mu} := \frac{1}{n}\sum_{i=1}^n \mu_i$, then we have*

$$\mathbb{E}_{\boldsymbol{x}_1\sim\mu_1, \cdots, \boldsymbol{x}_n\sim\mu_n}\left[\sup_{f\in\mathcal{F}}\left(\mathbb{E}_{\boldsymbol{x}\sim\bar{\mu}}[f(\boldsymbol{x})] - \frac{1}{n}\sum_{i=1}^n f(\boldsymbol{x}_i)\right)\right] \leq 2\mathcal{R}_n(\mathcal{F}).$$

*Proof.* The proof follows with the classical Rademacher complexity [74, Chapter 26] apart from the averaged measure. Take a copy of $X$, i.e., $X' = \{\boldsymbol{x}_i'\}_{i=1}^n$ such that $X'$ is independent but

$\boldsymbol{x}_i' \sim \mu_i, \forall i \in [n]$. According to Eq. (25), we have

$$\mathbb{E}_{\boldsymbol{x}\sim\bar{\mu}}[f(\boldsymbol{x})] = \mathbb{E}_{\boldsymbol{x}_1'\sim\mu_1,\cdots,\boldsymbol{x}_n'\sim\mu_n}\left[\frac{1}{n}\sum_{i=1}^n f(\boldsymbol{x}_i')\right]. \tag{27}$$

Note that every possible configuration/value of $\boldsymbol{\xi}$ has probability of $1/2^n$ due to $\boldsymbol{\xi} \in \{-1,1\}^n$. Without loss of generality, we can permute any configuration of $\boldsymbol{\xi}$ of such that

$$\xi_{u_1} = \xi_{u_2} = \cdots = \xi_{u_k} = 1, \quad \xi_{u_{k+1}} = \xi_{u_{k+2}} = \cdots = \xi_{u_n} = -1, \ k \in \{0\} \cup [n],$$

where $\boldsymbol{u} = \{u_1, u_2, \cdots, u_n\}$ is a permutation of $\{1, 2, \ldots, n\}$. Accordingly, for any configuration of $\boldsymbol{\xi}$, we have

$$\mathbb{E}_{\{\boldsymbol{x}_i\}_{i=1}^n}\left[\mathbb{E}_{\{\boldsymbol{x}_i'\}_{i=1}^n}\left[\sup_{f\in\mathcal{F}}\left(\frac{1}{n}\sum_{i=1}^n \xi_i\left(f\left(\boldsymbol{x}_i'\right) - f\left(\boldsymbol{x}_i\right)\right)\right)\right]\right]$$

$$= \mathbb{E}_{\{\boldsymbol{x}_i\}_{i=1}^n}\left[\mathbb{E}_{\{\boldsymbol{x}_i'\}_{i=1}^n}\left[\sup_{f\in\mathcal{F}}\left(\frac{1}{n}\left(\sum_{i=1}^k \left(f\left(\boldsymbol{x}_{u_i}'\right) - f\left(\boldsymbol{x}_{u_i}\right)\right) + \sum_{i=k+1}^n \left(f\left(\boldsymbol{x}_{u_i}\right) - f\left(\boldsymbol{x}_{u_i}'\right)\right)\right)\right)\right]\right]$$

$$= \mathbb{E}_{\{\boldsymbol{x}_i\}_{i=1}^n}\left[\mathbb{E}_{\{\boldsymbol{x}_i'\}_{i=1}^n}\left[\sup_{f\in\mathcal{F}}\left(\frac{1}{n}\sum_{i=1}^n \left(f\left(\boldsymbol{x}_i'\right) - f\left(\boldsymbol{x}_i\right)\right)\right)\right]\right], \tag{28}$$

where we use the fact that $\boldsymbol{x}_{u_i}$ and $\boldsymbol{x}_{u_i}'$ are independent and symmetric. Based on this, we obtain

$$\mathbb{E}_{\{\boldsymbol{x}_i\}_{i=1}^n}\left[\sup_{f\in\mathcal{F}}\left(\mathbb{E}_{\boldsymbol{x}\sim\bar{\mu}}[f(\boldsymbol{x})] - \frac{1}{n}\sum_{i=1}^n f\left(\boldsymbol{x}_i\right)\right)\right]$$

$$= \mathbb{E}_{\{\boldsymbol{x}_i\}_{i=1}^n}\left[\sup_{f\in\mathcal{F}}\left(\mathbb{E}_{\{\boldsymbol{x}_i'\}_{i=1}^n}\left[\frac{1}{n}\sum_{i=1}^n f\left(\boldsymbol{x}_i'\right)\right] - \frac{1}{n}\sum_{i=1}^n f\left(\boldsymbol{x}_i\right)\right)\right] \quad \text{[using Eq. (27)]}$$

$$= \mathbb{E}_{\{\boldsymbol{x}_i\}_{i=1}^n}\left[\sup_{f\in\mathcal{F}}\left(\mathbb{E}_{\{\boldsymbol{x}_i'\}_{i=1}^n}\left[\frac{1}{n}\sum_{i=1}^n f\left(\boldsymbol{x}_i'\right) - \frac{1}{n}\sum_{i=1}^n f\left(\boldsymbol{x}_i\right)\right]\right)\right]$$

$$\leqslant \mathbb{E}_{\{\boldsymbol{x}_i\}_{i=1}^n}\left[\mathbb{E}_{\{\boldsymbol{x}_i'\}_{i=1}^n}\left[\sup_{f\in\mathcal{F}}\left(\frac{1}{n}\sum_{i=1}^n f\left(\boldsymbol{x}_i'\right) - \frac{1}{n}\sum_{i=1}^n f\left(\boldsymbol{x}_i\right)\right)\right]\right] \quad \text{[Jensen's inequality]}$$

$$= \mathbb{E}_{\{\boldsymbol{x}_i\}_{i=1}^n}\left[\mathbb{E}_{\{\boldsymbol{x}_i'\}_{i=1}^n}\left[\mathbb{E}_{\boldsymbol{\xi}}\left[\sup_{f\in\mathcal{F}}\left(\frac{1}{n}\sum_{i=1}^n \xi_i\left(f\left(\boldsymbol{x}_i'\right) - f\left(\boldsymbol{x}_i\right)\right)\right)\right]\right]\right] \quad \text{[using Eq. (28)]}$$

$$\leqslant \mathbb{E}_{\{\boldsymbol{x}_i\}_{i=1}^n}\left[\mathbb{E}_{\boldsymbol{\xi}}\left[\sup_{f\in\mathcal{F}}\left(\frac{1}{n}\sum_{i=1}^n \xi_i f\left(\boldsymbol{x}_i\right)\right)\right]\right] + \mathbb{E}_{\{\boldsymbol{x}_i'\}_{i=1}^n}\left[\mathbb{E}_{\boldsymbol{\xi}}\left[\sup_{f\in\mathcal{F}}\left(\frac{1}{n}\sum_{i=1}^n \xi_i f\left(\boldsymbol{x}_i'\right)\right)\right]\right]$$

$$= 2\mathcal{R}_n(\mathcal{F}), \tag{29}$$

where the last inequality holds by the fact that $\xi_i$ and $-\xi_i$, $i \in [n]$ admit the same distribution, and multiplying each term in the summation by a Rademacher variable $\xi_i$ will not change the expectation due to $\mathbb{E}\xi_i = 0$.

$\square$

Based on the above lemma, we demonstrate that the Rademacher complexity can be well approximated by the empirical Rademacher complexity under our non-iid setting.

**Lemma 7.** *Under the same setting of Lemma 6, for any $f \in \mathcal{F}$, assume that $|f(\boldsymbol{x}) - f(\boldsymbol{x}')| \leqslant c, \ \forall \boldsymbol{x}, \boldsymbol{x}' \in \mathrm{dom}(f)$ for some constant $c > 0$, for any $\delta \in (0,1)$, the following proposition holds with probability at least $1 - \delta$*

$$\mathrm{Pr}\left(\mathbb{E}_{\boldsymbol{x}\sim\bar{\mu}}(f(\boldsymbol{x})) - \frac{1}{n}\sum_{i=1}^n f\left(\boldsymbol{x}_i\right) \geqslant 2\widehat{\mathcal{R}}_n(\mathcal{F}, X) + 3\delta\right) \leqslant 2\exp\left(-\frac{2n\delta^2}{c^2}\right). \tag{30}$$

*Proof.* The proof follows with the classical Rademacher complexity [74, Chapter 26] apart from the averaged measure. Recall the definition of the empirical Rademacher complexity in Eq. (26), $\widehat{\mathcal{R}}_n(\mathcal{F}, X)$ is a function of $n$ random variables $\{\boldsymbol{x}_i\}_{i=1}^n$. Moreover, due to $|f(\boldsymbol{x}) - f(\boldsymbol{x}')| \leqslant c$, $\widehat{\mathcal{R}}_n(\mathcal{F}, X)$ satisfies the precondition for McDiarmid's inequality by at most $c/n$, which only requires independence of random variables without the identically distributed condition

$$\Pr\left(\widehat{\mathcal{R}}_n(\mathcal{F}, X) - \mathbb{E}_{\{\boldsymbol{x}_i\}_{i=1}^n}[\widehat{\mathcal{R}}_n(\mathcal{F}, X)] \geqslant \delta\right) \leqslant \exp\left(-\frac{2n\delta^2}{c^2}\right),$$

which implies

$$\Pr\left(\left|\widehat{\mathcal{R}}_n(\mathcal{F}, X) - \mathcal{R}_n(\mathcal{F})\right| \geqslant \delta\right) \leqslant 2\exp\left(-\frac{2n\delta^2}{c^2}\right). \tag{31}$$

By Lemma 6, we have

$$\mathbb{E}_{\{\boldsymbol{x}_i\}_{i=1}^n}\left[\mathbb{E}_{\boldsymbol{x}\sim\bar{\mu}}[f(\boldsymbol{x})] - \frac{1}{n}\sum_{i=1}^n f(\boldsymbol{x}_i)\right] \leqslant \mathbb{E}_{\{\boldsymbol{x}_i\}_{i=1}^n}\left[\sup_{f\in\mathcal{F}}\left(\mathbb{E}_{\boldsymbol{x}\sim\bar{\mu}}[f(\boldsymbol{x})] - \frac{1}{n}\sum_{i=1}^n f(\boldsymbol{x}_i)\right)\right] \leqslant 2\mathcal{R}_n(\mathcal{F}).$$

Denote event A as

$$\left[\mathbb{E}_{\boldsymbol{x}\sim\bar{\mu}}[f(\boldsymbol{x})] - \frac{1}{n}\sum_{i=1}^n f(\boldsymbol{x}_i)\right] - \mathbb{E}_{\{\boldsymbol{x}_i\}_{i=1}^n}\left[\mathbb{E}_{\boldsymbol{x}\sim\bar{\mu}}[f(\boldsymbol{x})] - \frac{1}{n}\sum_{i=1}^n f(\boldsymbol{x}_i)\right] \geqslant \delta,$$

we use McDiarmid's inequality again to obtain $\Pr(\text{A}) \leqslant e^{-2n\delta^2/c^2}$ since $\mathbb{E}_{\boldsymbol{x}\sim\bar{\mu}}[f(\boldsymbol{x})] - \frac{1}{n}\sum_{i=1}^n f(\boldsymbol{x}_i)$ can be regarded as a function of $\{\boldsymbol{x}_i\}_{i=1}^n$ and any variations of $\{\boldsymbol{x}_i\}_{i=1}^n$ would change the outcome by at most $c/n$. Denote event B as $\mathbb{E}_{\boldsymbol{x}\sim\bar{\mu}}[f(\boldsymbol{x})] - \frac{1}{n}\sum_{i=1}^n f(\boldsymbol{x}_i) - 2\mathcal{R}_n(\mathcal{F}) \geqslant \delta$, we have $\Pr(\text{B}) \leqslant \Pr(\text{A}) \leqslant e^{-2n\delta^2/c^2}$.

Further, denote the event C as $\widehat{\mathcal{R}}_n(\mathcal{F}, X) \geqslant \mathcal{R}_n(\mathcal{F}) - \delta$, we have $\Pr(\text{C}) \geqslant 1 - \exp(-2n\delta^2/c^2)$ by Eq. (31). Denote the event D as $\mathbb{E}_{\boldsymbol{x}\sim\bar{\mu}}(f(\boldsymbol{x})) - \frac{1}{n}\sum_{i=1}^n f(\boldsymbol{x}_i) \geqslant 2\widehat{\mathcal{R}}_n(\mathcal{F}) + 3\delta$, we have

$$\Pr\left(\mathbb{E}_{\boldsymbol{z}\sim\bar{\mu}}(f(\boldsymbol{x})) - \frac{1}{n}\sum_{i=1}^n f(\boldsymbol{x}_i) \geqslant 2\widehat{\mathcal{R}}_n(\mathcal{F}) + 3\delta\right) = \Pr(\text{D}) = \Pr(\text{C}\cap\text{D}) + \Pr(\text{C}\cup\text{D}) - \Pr(\text{C})$$

$$\leqslant \Pr(\text{B}) + 1 - \Pr(\text{C})$$

$$= 2\exp\left(-2n\delta^2/c^2\right),$$

which concludes the proof.

$\square$

Similar to the proof of Lemma 7, it is easy to verify that, the standard Massart's lemma and the Talagrand's Contraction Lemma (empirical Rademacher complexity of Lipschitz function class) in [74, Chapter 26] are valid to our independent but non-iid setting.

## D.2  Local Rademacher complexity

Here we present some results on local Rademacher complexity [75] that is needed in this work. The used lemmas here are still valid for our independent but non-identically distributed data. Since the proof framework is similar to what we present for Rademacher complexity, we omit the proofs here.

When applying local Rademacher complexity, we need the following definition.

**Definition 5.** *A function $\psi : \mathbb{R}_+ \to \mathbb{R}_+$ is sub-root if it is non-negative, non-decreasing, and if $\psi(x)/\sqrt{x}$ is non-increasing.*

**Lemma 8.** *[76, Theorem 2] Let $\mathcal{F}$ be a function class with $\|f\|_\infty \leqslant b$, $\forall f \in \mathcal{F}$ and $\widetilde{F} := \{f - g : f, g \in \mathcal{F}\}$, and $P_n f := \frac{1}{n}\sum_{i=1}^n f(\boldsymbol{x}_i)$, then taking the average measure $\bar{\mu}$, we have*

$$\mathcal{R}_n\{f \in \mathcal{F} : \mathbb{E}_{\bar{\mu}} f^2 \leqslant R\} \leqslant \inf_{\epsilon>0}\left[2\mathcal{R}_n\{f \in \widetilde{\mathcal{F}} : P_n f^2 \leqslant \epsilon^2\} + \frac{8b\log\left(\epsilon/2, \mathcal{F}, \|\cdot\|_2\right)}{n}\right.$$

$$\left. + \sqrt{\frac{2R\log\mathscr{N}\left(\epsilon/2, \mathcal{F}, \|\cdot\|_2\right)}{n}}\right].$$

**Lemma 9.** *[75, Theorem 3.3, modified version] Let $f$ be a class of functions with ranges in $[a, b]$ and assume that there exists some functional $T : \mathcal{F} \to \mathbb{R}^+$ and some constant $B$ such that $\mathrm{Var}(f) \leqslant T(f) \leqslant BPf$ for every $f \in \mathcal{F}$. Let $P_n$ be the empirical measure supported on the independent data points $\{\boldsymbol{x}_i\}_{i=1}^n$ with the averaged measure $\bar{\mu} := \frac{1}{n}\mu_i$, Let $\psi$ be a sub-root function with the fixed point $R^*$. If for any $R \geqslant R^*$, $\psi$ satisfies*

$$\psi(R) \geqslant B\mathcal{R}_n\{f \in \mathcal{F} : T(f) \leqslant R\},$$

*then for any $J > 1$ and $\delta \in (0, 1)$, with probability at least $1 - \delta$, we have*

$$\mathbb{E}_{\bar{\mu}} f \leqslant \frac{J}{J-1}P_n f + \frac{c_1 J}{B} R^* + (c_2(b-a) + c_3 BJ)\frac{\log(1/\delta)}{n},$$

*where $c_1$, $c_2$, $c_3$ are some positive constants.*

**Lemma 10.** *[73, Refined entropy integral] Let $P_n$ be the empirical measure supported on the independent data points $\{\boldsymbol{x}_i\}_{i=1}^n$. For any function class $\mathcal{F}$ and any monotone sequence $\{\epsilon_k\}_{k=0}^\infty$ decreasing to 0 such that $\epsilon_0 \geqslant \sup_{f \in \mathcal{F}} \sqrt{P_n f^2}$, the following inequality holds for every non-negative integer $N$*

$$\widehat{\mathcal{R}}_n(\mathcal{F}, X) \leqslant 4\sum_{k=1}^N \epsilon_{k-1}\sqrt{\frac{\log \mathcal{N}(\epsilon_k, \mathcal{F}, \|\cdot\|_2)}{n}} + \epsilon_N. \tag{32}$$

# E Proofs of regret bounds via deep ReLU neural networks

In this section, we give the proofs of regret bounds via deep ReLU neural networks according to the function class of $\mathbb{T}_h^\star Q$ in Besov spaces.

To conclude our proof, we need the following lemma that how well the functions in the Besov space can be approximated by deep neural networks with the ReLU activation. Here the approximation error is defined in the $L^4(\mathcal{X})$-integrable space (*c.f.* Corollary 1).

**Lemma 11.** *(Approximation error in Besov space) [30, Proposition 1, modified version] Assume that the smoothness parameter $\alpha$ satisfies*

$$\alpha > \eta := d(1/p - 1/4)_+,$$

*then there exists a deep neural network architecture $\mathcal{F}_{\mathrm{DNN}}(L, m, S, B)$ with $\nu := (\alpha - \eta)/(2\eta)$ and a large $N$ such that*

$$L \asymp \log N, \quad S \asymp N, \quad m \asymp N \log N, \quad and \quad B \asymp N^{1/\nu + 1/d}, \tag{33}$$

*then it holds that*

$$\sup_{f^* \in \mathcal{B}_{p,q}^\alpha(\mathcal{X})} \inf_{f \in \mathcal{F}_{\mathrm{DNN}}(L,m,S,B)} \|f - f^*\|_{L^4(\mathcal{X})} \lesssim N^{-\frac{\alpha}{d}}, \qquad \forall q > 0.$$

In our proof, we need the following result on local Rademacher complexity of deep ReLU neural networks.

**Lemma 12.** *Let $X = \{\boldsymbol{x}_i\}_{i=1}^n \subseteq [0, 1]^d$ be an independent but non-identical distributed data set with $\boldsymbol{x}_i \sim \mu_i, \forall i \in [n]$, and $\mathcal{R}_n\{f \in \mathcal{F}_{\mathrm{DNN}} : Pf^2 \leqslant R\}$ be the local Rademacher complexity of the function class $\mathcal{F}_{\mathrm{DNN}}$ on $X$ defined in Eq. (4), denote the averaged measure as $\bar{\mu} := \frac{1}{n}\sum_{i=1}^n \mu_i$, then for a large $N$, we have*

$$\mathcal{R}_n\{f \in \mathcal{F}_{\mathrm{DNN}} : \mathbb{E}_{\bar{\mu}} f^2 \leqslant R\} \lesssim \left(\frac{1}{n} + \sqrt{\frac{R}{n}}\right)\sqrt{N[(\log N)^2 + \log n]} + \frac{HN[(\log N)^2 + \log n]}{n}. \tag{34}$$

**Remark:** The parameter $N$ depends on the number of the training data $n$, but it will be determined later.

*Proof.* According to [30, Lemma 3], the covering number of $\mathcal{F}_{\mathrm{DNN}}$ can be bounded by

$$\log \mathcal{N}(\varepsilon, \mathcal{F}_{\mathrm{DNN}}, \|\cdot\|_2) \leqslant \log \mathcal{N}(\varepsilon, \mathcal{F}_{\mathrm{DNN}}, \|\cdot\|_\infty) \leq 2SL\log\left(\frac{L(B \vee 1)(m+1)}{\varepsilon}\right)$$

$$\lesssim N\left[(\log N)^2 + \log\left(\frac{1}{\varepsilon}\right)\right].$$

Denote $\widetilde{\mathcal{F}}_{\text{DNN}} = \{f - g : f, g \in \mathcal{F}_{\text{DNN}}\}$, it satisfies

$$
\log \mathcal{N}(\varepsilon, \widetilde{\mathcal{F}}_{\text{DNN}}, \|\cdot\|_2) \leqslant 2 \log \mathcal{N}\left(\frac{\varepsilon}{2}, \mathcal{F}_{\text{DNN}}, \|\cdot\|_2\right) \leqslant 2 \log \mathcal{N}\left(\frac{\varepsilon}{2}, \mathcal{F}_{\text{DNN}}, \|\cdot\|_\infty\right)
$$

$$
\lesssim N\left[(\log N)^2 + \log\left(\frac{2}{\varepsilon}\right)\right]. \tag{35}
$$

According to Lemma 10, taking $\varepsilon_j = 2^{-j}\varepsilon$, and using the inequality

$$
\mathcal{N}(\varepsilon_j, \{f \in \widetilde{\mathcal{F}}_{\text{DNN}} : P_n f^2 \leqslant \varepsilon^2\}, \|\cdot\|_2) \leqslant \mathcal{N}(\varepsilon_j/2, \widetilde{\mathcal{F}}_{\text{DNN}}, \|\cdot\|_2),
$$

then the following inequality holds for any $J \in \mathbb{N}^+$:

$$
\begin{aligned}
\mathcal{R}_n\{f \in \widetilde{\mathcal{F}}_{\text{DNN}} : P_n f^2 \leqslant \varepsilon^2\} &= \mathbb{E}\widehat{\mathcal{R}}_n\{f \in \widetilde{\mathcal{F}}_{\text{DNN}} : P_n f^2 \leqslant \varepsilon^2\} \\
&\leqslant 4\mathbb{E}\sum_{j=1}^{J} \varepsilon_{j-1}\sqrt{\frac{\log \mathcal{N}(\varepsilon_j/2, \widetilde{\mathcal{F}}_{\text{DNN}}, \|\cdot\|_2)}{n}} + \varepsilon_J \\
&\leqslant 4\mathbb{E}\sum_{j=1}^{J} 2^{-(j-1)}\varepsilon\sqrt{\frac{2\log \mathcal{N}(\frac{\varepsilon}{2^{(j+1)}}, \widetilde{\mathcal{F}}_{\text{DNN}}, \|\cdot\|_\infty)}{n}} + \varepsilon_J \\
&\lesssim \frac{\varepsilon}{\sqrt{n}}\sum_{j=1}^{J} 2^{-(j-1)}\sqrt{2N\left[(\log N)^2 + \log\left(\frac{2^{j+1}}{\varepsilon}\right)\right]} + 2^{-J}\varepsilon \\
&\lesssim \frac{\varepsilon}{\sqrt{n}}\sqrt{N\left[(\log N)^2 + \log\left(\frac{1}{\varepsilon}\right)\right]}, \quad [\text{taking } J \to \infty]
\end{aligned} \tag{36}
$$

where the first inequality holds by Lemma 10 and the second and third inequalities hold by Eq. (35). The last inequality uses the fact that $\sum_{j=0}^{\infty} \frac{\sqrt{j+1}}{2^{j-1}} < \infty$.

According to Lemma 8 with $\sup_{f \in \mathcal{F}_{\text{DNN}}} \|f\|_\infty \leqslant H$, we have

$$
\begin{aligned}
\mathcal{R}_n\{f \in \mathcal{F}_{\text{DNN}} : Pf^2 \leqslant R\} &\lesssim \inf_{\varepsilon > 0}\Bigg[2\mathbb{E}\mathcal{R}_n\{f \in \widetilde{\mathcal{F}}_{\text{DNN}} : P_n f^2 \leqslant \varepsilon^2\} \\
&\qquad + \frac{8HN\left[(\log N)^2 + \log\left(\frac{1}{\varepsilon}\right)\right]}{n} + \sqrt{\frac{2rN\left[\log^2 N + \log\left(\frac{1}{\varepsilon}\right)\right]}{n}}\Bigg] \\
&\lesssim \inf_{\varepsilon > 0}\Bigg[\frac{\epsilon + \sqrt{2R}}{\sqrt{n}}\sqrt{N\left[(\log N)^2 + \log\left(\frac{1}{\varepsilon}\right)\right]} + \frac{HN\left[\log^2 N + \log\left(\frac{1}{\varepsilon}\right)\right]}{n}\Bigg] \\
&\lesssim \frac{n^{-1/2} + \sqrt{R}}{\sqrt{n}}\sqrt{N\left(\log^2 N + \log n\right)} + \frac{HN\left(\log^2 N + \log n\right)}{n} := \psi(R),
\end{aligned} \tag{37}
$$

where we choose $\varepsilon := n^{-1/2}$ in the last inequality, and then we conclude the proof. $\qquad\square$

Based on the above result, we have the following proposition on generalization bounds in Besov spaces under non-iid state-action pairs.

**Proposition 2.** *Given the solution $\widehat{Q}_h^t = \arg\min_{f \in \mathcal{F}_{\text{DNN}}} \widehat{\mathcal{E}}_h^t(f)$ in Eq. (5), then for a large $N$ and any $\delta \in (0,1)$, with probability at least $1 - \delta$, we have*

$$
\mathcal{E}_h^t(\widehat{Q}_h^t) - \min_{f \in \mathcal{F}_{\text{DNN}}} \mathcal{E}_h^t(f) \lesssim \frac{N\left[(\log N)^2 + \log n\right]}{n} + \frac{H\sqrt{N\left[(\log N)^2 + \log n\right]}}{n} + \frac{H^2 \log(1/\delta)}{n},
$$

*where $n := \tilde{t}$ in our RL setting and $N$ depends on $t$ which needs further determined.*

*Proof.* It is clear that $\psi(R)$ defined in Eq. (37) in Lemma 12 is a sub-root function. Therefore, the fixed point $R^*$ of $\psi(R)$ can be analytically solved by the equation $R^* = \psi(R^*)$, which leads to

$$R^* \lesssim \frac{\sqrt{N\left[(\log N)^2 + \log n\right]}}{n} + \frac{HN\left[(\log N)^2 + \log n\right]}{n}.$$

Strictly speaking, there is an extra term $N\left[(\log N)^2 + \log n\right]^{\frac{3}{4}}/n$ in the above equation, but we can omit it as we only concern the smallest and largest order. By verifying the variance-expectation condition, we have

$$\mathbb{E}[\mathcal{E}_h^t(\widehat{Q}_h^t) - \mathcal{E}_h^t(f_h^\star)]^2 \leqslant 16H^2\mathbb{E}[\mathcal{E}_h^t(\widehat{Q}_h^t) - \mathcal{E}_h^t(f_h^\star)], \tag{38}$$

where $f_h^\star := \operatorname{argmin}_{f\in\mathcal{F}_{\text{DNN}}} \mathcal{E}_h^t(f)$ and we use the fact $\mathcal{E}_h^t(f)$ is $4H$-Lipschitz. Denote the function space $\widehat{\mathcal{F}_{\text{DNN}}}$ with the following function formulation for any $j \in [n]$

$$\hat{g}_h^t := \left[\widehat{Q}_h^t(s_h^{\tau_j}, a_h^{\tau_j}) - r_h(s_h^{\tau_j}, a_h^{\tau_j}) - V_{h+1}^t(s_{h+1}^{\tau_j})\right]^2 - \left[f_h^\star(s_h^{\tau_j}, a_h^{\tau_j}) - r_h(s_h^{\tau_j}, a_h^{\tau_j}) - V_{h+1}^t(s_{h+1}^{\tau_j})\right]^2,$$

we have $P_n\hat{g}_h^t = \widehat{\mathcal{E}}_h^t(\widehat{Q}_h^t) - \widehat{\mathcal{E}}_h^t(f_h^\star) \leqslant 0$ due to $\widehat{Q}_h^t = \operatorname{argmin}_{f\in\mathcal{F}} \widehat{\mathcal{E}}_h^t(f)$. Then using $\mathbb{E}g^2 \leqslant H^2Pg$, for any $g \in \widehat{\mathcal{F}_{\text{DNN}}}$ by Eq. (38), according to Lemma 9, the following inequality holds with probability at least $1 - \delta$

$$P\hat{g}_h^t \lesssim \frac{J}{H^2}R^* + \frac{(H^2J + H)\log(1/\delta)}{n}, \quad \forall J > 1,$$

where which further implies

$$\mathcal{E}_h^t(\widehat{Q}_h^t) - \min_{f\in\mathcal{F}_{\text{DNN}}} \mathcal{E}_h^t(f) \lesssim \frac{N\left[(\log N)^2 + \log n\right]}{n} + \frac{H\sqrt{N\left[(\log N)^2 + \log n\right]}}{n} + \frac{H^2\log(1/\delta)}{n}.$$

Finally, we conclude the proof. $\qquad\square$

*Proof of Theorem 1.* Using the approximation error in $L^4(\mathcal{X})$ by Corollary 1, the smoothness parameter satisfies $\alpha > d(1/p - 1/4)_+$. By taking $\delta/2$ in Proposition 2, we have

$$\|\Gamma_h^t\|_{L^2(\mathrm{d}\bar{\mu}_h^{\tilde{t}})}^2 \lesssim \left[\mathcal{E}_h^t(\widehat{Q}_h^t) - \min_{f\in\mathcal{F}_{\text{DNN}}} \mathcal{E}_h^t(f)\right] + \inf_{f\in\mathcal{F}_{\text{DNN}}} \|f - \mathbb{T}_h^\star Q_{h+1}^t\|_{L^4(\mathcal{X})}^2 \quad \text{[using Corollary 1]}$$

$$\lesssim N^{-\frac{2\alpha}{d}} + \frac{N\left[(\log N)^2 + \log \tilde{t}\right]}{\tilde{t}} + \frac{H\sqrt{N\left[(\log N)^2 + \log \tilde{t}\right]}}{\tilde{t}} + \frac{H^2\log(2/\delta)}{\tilde{t}}, \tag{39}$$

where in the second inequality, taking $\alpha > d(1/p - 1/4)_+$, the approximation error can be estimated by Lemma 11

$$\inf_{f\in\mathcal{F}_{\text{DNN}}} \|f - \mathbb{T}_h^\star Q_{h+1}^t\|_{L^4(\mathcal{X})}^2 \lesssim N^{-2\alpha/d}.$$

Accordingly, the right hand side of Eq. (39) can be minimized by taking $N \asymp \tilde{t}^{\frac{d}{2\alpha+d}}$ up to $(\log \tilde{t})^3$-order in Eq. (33) for choosing suitable $L, m, S, B$. To make the architecture of deep RL independent of a variable $\tilde{t}$ (or $t$) during different episodes, here we directly choose $N \asymp T^{\frac{d}{2\alpha+d}}\log^3 T$, in this case, Eq. (39) can be formulated as

$$\|\Gamma_h^t\|_{L^2(\mathrm{d}\bar{\mu}_h^{\tilde{t}})}^2 \lesssim HT^{-\frac{2\alpha}{2\alpha+d}}\log^3\tilde{t} + \frac{T^{\frac{d}{2\alpha+d}}\log^5 T}{\tilde{t}} + \frac{H^2\log(2/\delta)}{\tilde{t}},$$

which requires the depth $L$ and the width $m$ up to

$$L \asymp \frac{d}{2\alpha+d}\log T, \quad m \asymp \frac{d}{2\alpha+d}T^{\frac{d}{2\alpha+d}}\log T.$$

Recall $\tilde{t} := \lceil \varrho t \rceil$, according to Lemma 4, if $\alpha > d(1/p - 1/4)_+$, then for any $\delta \in (0,1)$, with probability at least $1 - \delta/2$, the $\texttt{Term(i)}$ can be upper bounded by

$$
\begin{aligned}
\texttt{Term(i)} &\lesssim \left(\frac{\epsilon}{A}\right)^{-\frac{K}{2}} H\sqrt{T} \sqrt{\sum_{t=1}^{T} T^{-\frac{2\alpha}{2\alpha+d}} \log^3 \varrho t + \frac{T^{\frac{d}{2\alpha+d}} \log^5 T}{\varrho t} + \frac{H^2 \log(2/\delta)}{\varrho t}} + H\sqrt{T} \\
&\lesssim \left(\frac{\epsilon}{A}\right)^{-\frac{K}{2}} H\sqrt{T} \sqrt{HT^{\frac{d}{2\alpha+d}} \log^3 T + \frac{1}{\varrho} \int_1^{T+1} \left(\frac{T^{\frac{d}{2\alpha+d}} \log^5 T}{t} + \frac{H^2 \log(2/\delta)}{t}\right) \mathrm{d}t} + H\sqrt{T} \\
&\lesssim \left(\frac{\epsilon}{A}\right)^{-\frac{K}{2}} \frac{1}{\sqrt{\varrho}} \left(\sqrt{TH^3}\sqrt{T^{\frac{d}{2\alpha+d}} \log^6 T} + H^2\sqrt{T}\sqrt{\log(2/\delta)\log T}\right) + H\sqrt{T} \\
&\lesssim \left(\frac{\epsilon}{A}\right)^{-\frac{K}{2}} \frac{1}{\sqrt{\varrho}} \left(H^{\frac{3}{2}} T^{\frac{\alpha+d}{2\alpha+d}} \log^3 T + H^2\sqrt{T}\sqrt{\log(2/\delta)}\log T\right) + H\sqrt{T} .
\end{aligned}
\tag{40}
$$

Then taking $\delta/2$ in the statistical error $\texttt{Term(ii)}$ in Lemma 1, if $\alpha > d(1/p - 1/4)_+$, with probability at least $1 - \delta$, we have

$$
\text{Regret}(T) \lesssim \left(\frac{\epsilon}{A}\right)^{-\frac{K}{2}} \frac{1}{\sqrt{\varrho}} \left(H^{\frac{3}{2}} T^{\frac{\alpha+d}{2\alpha+d}} \log^3 T + H^2\sqrt{T}\sqrt{\log\left(\frac{2}{\delta}\right)}\log T\right) + \epsilon HT + \sqrt{TH^3 \log\left(\frac{4}{\delta}\right)} .
$$

Then taking

$$
\epsilon = \mathcal{O}\left(H^{\frac{2}{K+2}} K^{\frac{2}{K+2}} A^{\frac{K}{K+2}} T^{-\frac{2\alpha}{(2\alpha+d)(K+2)}}\right),
$$

which implies

$$
\text{Regret} \lesssim \widetilde{\mathcal{O}}\left(H^{\frac{H+4}{H+2}} K^{\frac{2}{K+2}} A^{\frac{K}{K+2}} T^{\frac{\alpha K + (\alpha+d)(K+2)}{(2\alpha+d)(K+2)}}\right) .
$$

Finally we conclude the proof.

$\square$

## F Proofs of regret bounds via two-layer neural networks

In this section, we focus on generalization bounds under the independent but non-identically distributed data setting in the Barron space, and it is useful to present estimates of our regret bound.

**Lemma 13.** *For two-layer ReLU neural networks with bounded $\ell_1$ path norm defined in Eq. (3) given the function class $\mathcal{F}_{\mathrm{SNN}}$ and $n$ independent but non-identically distributed data points $X = \{\boldsymbol{x}_i\}_{i=1}^{n} \subseteq \mathcal{X}$, then we have*

$$
\mathcal{R}_n(\mathcal{F}_{\mathrm{SNN}}) \leqslant 2B\sqrt{\frac{2\log(2d)}{n}} .
$$

*Proof.* Here we directly focus on the $\ell_1$ path norm, which is different from [31, Theorem 3]. Based on the definition of two-layer ReLU neural networks defined in Eq. (3), denote $\widetilde{\boldsymbol{w}}_k := (\boldsymbol{w}_k^\top, c_k)^\top$ and $\widetilde{\boldsymbol{x}} = (\boldsymbol{x}^\top, 1)^\top$ for simplicity, the empirical Rademacher complexity of $\mathcal{F}_{\mathrm{SNN}}$ under our setting can be

upper bounded by

$$
\begin{aligned}
\widehat{\mathcal{R}}_n(\mathcal{F}_{\text{SNN}}, X) &= \frac{1}{n} \mathbb{E}_{\boldsymbol{\xi}} \left[ \sup_{f \in \mathcal{F}_{\text{SNN}}} \frac{1}{m} \sum_{k=1}^{m} b_k \sum_{i=1}^{n} \xi_i \sigma(\widetilde{\boldsymbol{w}}_k^\top \widetilde{\boldsymbol{x}}) \right] \\
&\leqslant \mathbb{E}_{\boldsymbol{\xi}} \left[ \sup_{f \in \mathcal{F}_{\text{SNN}}} \frac{1}{m} \sum_{k=1}^{m} |b_k| \, \|\widetilde{\boldsymbol{w}}_k\|_1 \frac{1}{n} \left| \sum_{i=1}^{n} \xi_i \sigma \left( \frac{\widetilde{\boldsymbol{w}}_i}{\|\widetilde{\boldsymbol{w}}_i^\top\|_1} \widetilde{\boldsymbol{x}}_i \right) \right| \right] \\
&\leq B \mathbb{E}_{\boldsymbol{\xi}} \left[ \sup_{\|\widetilde{\boldsymbol{w}}\|_1 \leqslant 1} \frac{1}{n} \left| \sum_{i=1}^{n} \xi_i \sigma(\widetilde{\boldsymbol{w}}^\top \widetilde{\boldsymbol{x}}_i) \right| \right] \\
&\leqslant 2B \mathbb{E}_{\boldsymbol{\xi}} \left[ \sup_{\|\widetilde{\boldsymbol{w}}\|_1 \leqslant 1} \frac{1}{n} \sum_{i=1}^{n} \xi_i \widetilde{\boldsymbol{w}}^\top \widetilde{\boldsymbol{x}}_i \right] \quad \text{[using symmetry of } \boldsymbol{\xi} \text{ and 1-Lipschitz of ReLU]} \\
&\leqslant 2B \mathbb{E}_{\boldsymbol{\xi}} \left\| \frac{1}{n} \sum_{i=1}^{n} \xi_i \widetilde{\boldsymbol{x}}_i \right\|_\infty \quad , \quad \text{[using Hölder inequality]}
\end{aligned}
$$

where the first inequality holds by the homogeneity of ReLU for any $\widetilde{\boldsymbol{w}} \in \mathbb{R}^d / \{\boldsymbol{0}\}$. Since the Massart's lemma is still valid under our independent but non-identically distributed data, $\mathcal{R}_n(\mathcal{F}_{\text{SNN}})$ can be further expressed by

$$
\widehat{\mathcal{R}}_n(\mathcal{F}_{\text{SNN}}, X) \leqslant 2B \mathbb{E}_{\boldsymbol{\xi}} \left\| \frac{1}{n} \sum_{i=1}^{n} \xi_i \widetilde{\boldsymbol{x}}_i \right\|_\infty \leqslant 2B \sqrt{2 \log(2d)/n} \,,
$$

where the last inequality holds by the maximum of $n$ sub-Gaussian random variables [72] since Rademacher random variables are sub-Gaussian, and finally we conclude the proof. □

*Proof of Theorem 2.* Denote $X := \{(s_h^{\tau_j}, a_h^{\tau_j}, s_{h+1}^{\tau_j})\}_{j=1}^{\tilde{t}}$ for simplicity and notice that the function $[f(s_h, a_h) - r_h(s_h, a_h) - V_{h+1}^t(s_{h+1})]^2$ is $4H$-Lipschitz. Then according to Lemma 7, for any $\delta \in (0, 1)$, the following result holds with probability at least $1 - \delta/2$

$$
\begin{aligned}
\widehat{\mathcal{E}}_h^t(f) - \mathcal{E}_h^t(f) &\leqslant 2\widehat{\mathcal{R}}_{t-1}(\mathcal{F}_{\text{SNN}}, X) + 12H \sqrt{\frac{\log(4/\delta)}{2\tilde{t}}} \\
&\leqslant 8B\widetilde{R}H \sqrt{\frac{2\log(2d)}{\tilde{t}}} + 12H \sqrt{\frac{\log(4/\delta)}{2\tilde{t}}} \,,
\end{aligned} \tag{41}
$$

where we use the empirical Rademacher complexity in Lemma 13. Accordingly, by Lemma 5 and Eq. (41), then with probability at least $1 - \delta/2$, we have

$$
\begin{aligned}
\|\Gamma_h^t\|_{L^2(\mathrm{d}\mu_h^{\tilde{t}})}^2 &\leqslant \left[ \mathcal{E}_h(\widehat{Q}_h^t) - \min_{f \in \mathcal{F}_{\text{SNN}}} \mathcal{E}_h(f) \right] + \inf_{f \in \mathcal{F}_{\text{SNN}}} \|f - \mathbb{T}_h^\star Q_{h+1}^t\|_{L^2(\mathrm{d}\bar{\mu}_h^{\tilde{t}})}^2 \\
&\leqslant \mathcal{E}_h(\widehat{Q}_h^t) - \min_{f \in \mathcal{F}_{\text{SNN}}} \mathcal{E}_h(f) + \frac{3\|\mathbb{T}_h^\star Q_{h+1}^t\|_{\mathcal{P}}^2}{m} \\
&\leqslant 2 \sup_{f \in \mathcal{F}_{\text{SNN}}} |\mathcal{E}_h^t(f) - \widehat{\mathcal{E}}_h^t(f)| + \frac{3\widetilde{R}^2}{m} \quad \text{[using Assumption 2]} \\
&\leqslant 16B\widetilde{R}H \sqrt{\frac{2\log(2d)}{\tilde{t}}} + 24H \sqrt{\frac{\log(4/\delta)}{2\tilde{t}}} + \frac{3\widetilde{R}^2}{m} \,, \quad \text{[using Eq. (41)]}
\end{aligned} \tag{42}
$$

where the second inequality uses the approximation result for two-layer ReLU neural networks and the Barron space in [31, Theorem 4]. Accordingly, by Lemma 4, for any $\delta \in (0, 1)$, the `Term(i)` in

the regret decomposition can be upper bounded with probability at least $1 - \delta/2$

$$\texttt{Term(i)} \lesssim \left(\frac{\epsilon}{A}\right)^{-\frac{K}{2}} H\sqrt{T} \sqrt{\sum_{t=1}^{T} \left(BH^2(\varrho t)^{-\frac{1}{2}}\sqrt{\log d} + H(\varrho t)^{-\frac{1}{2}}\sqrt{\log \frac{4}{\delta}} + \frac{H^2}{m}\right)} + H\sqrt{T}$$

$$\lesssim \left(\frac{\epsilon}{A}\right)^{-\frac{K}{2}} H\sqrt{T} \sqrt{\frac{1}{\sqrt{\varrho}} \int_{1}^{T+1} \left(BH^2 t^{-\frac{1}{2}}\sqrt{\log d} + Ht^{-\frac{1}{2}}\sqrt{\log \frac{4}{\delta}}\right) \mathrm{d}t + \frac{H^2 T}{m}} + H\sqrt{T}$$

$$\lesssim \left(\frac{\epsilon}{A}\right)^{-\frac{K}{2}} \frac{1}{\sqrt{\varrho}} \left[T^{\frac{3}{4}} H^2 B(\log d)^{\frac{1}{4}} + T^{\frac{3}{4}} H^{\frac{3}{2}} \log^{\frac{1}{4}}\left(\frac{4}{\delta}\right)\right] + \left(\frac{\epsilon}{A}\right)^{-\frac{K}{2}} \frac{H^2 T}{\sqrt{m}} + H\sqrt{T}.$$

$$\tag{43}$$

where we use $\widetilde{R} \asymp H$ and $\int_{1}^{T}(t-1)^{-1/2}\mathrm{d}t = \mathcal{O}(\sqrt{T})$.

Accordingly, taking $\delta/2$ in the statistical error $\texttt{Term(ii)}$ in Lemma 1, then with the probability at least $1 - \delta$, the total regret can be upper bounded by

$$\mathrm{Regret}(T) \lesssim \left(\frac{\epsilon}{A}\right)^{-\frac{K}{2}} \left(\frac{H^2 T^{\frac{3}{4}}}{\sqrt{\varrho}}\left[B(\log d)^{\frac{1}{4}} + \log^{\frac{1}{4}}\left(\frac{4}{\delta}\right)\right] + \frac{H^2 T}{\sqrt{m}}\right) + \epsilon H T + \sqrt{TH^3 \log\left(\frac{4}{\delta}\right)}.$$

Taking $m = \Omega(\sqrt{T})$ and $\epsilon = \mathcal{O}\left(H^{\frac{2}{K+2}} T^{-\frac{1}{2(K+2)}}\right)$, the regret bound can be further represented as

$$\mathrm{Regret}(T) \lesssim \widetilde{\mathcal{O}}(H^{\frac{K+4}{K+2}} T^{\frac{2K+3}{2K+4}}),$$

which concludes the proof. $\qquad\qquad\qquad\qquad\qquad\qquad\qquad\qquad\qquad\qquad\qquad\qquad\square$