# OpenReview forum: "Understanding Deep Neural Function Approximation in Reinforcement Learning via $\epsilon$-Greedy Exploration"
_NeurIPS.cc/2022/Conference — NeurIPS 2022 Accept_

### Official Review · Reviewer_Pwx6 · 2022-07-05

**Rating:** 6
**Confidence:** 3
**Soundness:** 3 good
**Presentation:** 2 fair
**Contribution:** 2 fair

**Summary:**

This paper studies a deep Q-network algorithm in the online and episodic MDP setting. With the probabilistic sufficient exploration assumption, this paper focuses on Besov and Barron function spaces to model the deep Q-networks. Two main theoretical guarantees are provided: one is for shadow neural networks, and the other is for deep neural networks. In both two cases, sublinear regret is achieved. From the theoretical results, this paper claims that for low smooth Q-functions, the width has a more important role and a constant depth is enough. From this perspective, this paper may shed light on width and depth guidelines in deep Q-learning.

===

After rebuttal, my concerns are addressed. This paper may shed new light on understanding deep neural networks in reinforcement learning. Yet, there are several future works, as noted in the conclusion part. Therefore, I change my score from 4 to 6.

**Questions:**

I have a few concerns about this paper's motivation and theoretical implications. I list my questions as follows.

1. Can the authors justify the (technical) challenge after Assumption 1 is satisfied?

I have this question because I believe how to address the exploration-and-exploitation trade-off is the core of online learning. However, if we make the so-called Probabilistic sufficient exploration, it is not clear (at least for me) what is challenging in this setting. It seems that it is sufficient to consider the offline setting, where a fixed dataset is provided, to study how to approximate optimal Q-value functions. I guess that the online setting considered in this paper may introduce unnecessary (technical) challenges. I would appreciate it if the authors could help clarify this point.

2. Can the authors further explain the empirical objective in Equation (3)?

It is known that DQN has two key components: experience replay and the target network. Without the target network, Q-learning with function approximation can easily diverge in theory; see [2] and Section 11.3 of [3]. From the workshop paper [4] to the nature version, the target network is introduced, and the performance is improved a lot for DQN. In practice, if the target network is removed, it is easy to see that DQN cannot work for games like Pong and Breakout.

This paper shows that i.i.d. sampling by experience replay is not necessary. This is acceptable. Then, this paper further studies Algorithm 1, where the target network is not used. This is a huge difference with practice. With a powerful approximation power, it is not very surprising that Algorithm 1 can converge. However, the issue is that the computation cost of Algorithm 1 is huge, which restricts the application of Algorithm 1. Based on this thought, I am not convinced that the theoretical implications in this paper are well-aligned with practical observations.  Please also refer to [5] for the study of online target Q-learning (authors are excused for this paper because this paper is recently posted).

[2] Baird, Leemon. "Residual algorithms: Reinforcement learning with function approximation." *Machine Learning Proceedings 1995*. Morgan Kaufmann, 1995. 30-37.

[3] Sutton, Richard S., and Andrew G. Barto. *Reinforcement learning: An introduction*. MIT press, 2018.

[4] Mnih, Volodymyr, et al. "Playing atari with deep reinforcement learning." *arXiv preprint arXiv:1312.5602* (2013).

[5] Zanette, Andrea, and Martin J. Wainwright. "Stabilizing Q-learning with Linear Architectures for Provably Efficient Learning." arXiv preprint arXiv:2206.00796 (2022).

**Limitations:**

This paper does a great job of analyzing Q-learning with deep neural networks. The main limitations are listed below.

- Assumption 1 is quite strong for studying online learning. Though this paper explains that epsilon-greedy can achieve this, it is known that epsilon-greedy is much inefficient.
- A gap between Algorithm 1 and practical one (i.e., the target network is not considered). It is hard for me to believe the theoretical implications in this paper are well-aligned with practical observations. If the authors can provide some numerical results to justify the theoretical results, I would be much convinced.

I would suggest that the authors can elaborate on the technical challenge and provide more justification for the setting studied in this paper.

**Minor Issues**

This writing of this paper can be further improved, as there are several grammar errors. To name a few,

- In Line 35, demonstrates -> demonstrate
- In Line 82, effects -> affects
- In Line 93, assuming a simulator can require any state and action -> assuming a simulator in which the agent can require any state and action
- In Line 99, the above algorithms -> the above bounds
- In Line 112, function spaces of two -> and function spaces of two
- In Line 177, which not heavily depends on -> which does not heavily depend on
- In Line 190, $\ell_1$ path norm -> $\ell_1$-path norm
- In Line 206, it is not clear the meaning of "this requirement"
- In Line 251, they -> [12]



**Strengths And Weaknesses:**

**Strengths**

- More border function classes are considered. Most prior works focus on the linear function approximation or (limited) non-linear function approximation.
- A tighter bound in terms of the dependence on the width and depth is provided. For the prior work that relies on NTK, an impractical width $m = \Omega(T^{13}) $is required.

**Weakness**

- The whole theory heavily relies on the assumption of probabilistic sufficient exploration. With this assumption, the exploration issue is already addressed. To this end, it is unclear why this paper studies the function approximation in the online setting. It seems that the offline setting considered in [1] is sufficient to study how to approximate optimal Q-value functions.
- To my knowledge, the value iteration algorithm (with neural networks) is rarely used in practice, meaning that a gap separates the theory in this paper from practical algorithms. To clarify, Algorithm 1 has two computational issues. First, it is hard to compute the global minimizer defined in the empirical objective (i.e., Equation (3)) for each iteration. The computation cost is unacceptable. Therefore, people prefer iterative algorithms like Q-learning. Second, Q-learning with function approximation can easily diverge, and the target network is introduced to address this issue in DQN. However, these features are not captured by Algorithm 1 and theoretical results.

[1] Jianqing Fan, Zhaoran Wang, Yuchen Xie, and Zhuoran Yang. A theoretical analysis of deep q-learning. In Learning for Dynamics and Control, pages 486–489, 2020.

---

> ### Author Response · Authors · 2022-08-01
> **Reviewer Pwx6 (Part I)**
>
> Thanks for your constructive comments and we prepare some clarifications on them. We have polished this paper according to your suggestions, highlighted in red for convenience.
>
> **Q1**: It is unclear why this paper studies the function approximation in the online setting. It seems that the offline setting considered in [1] is sufficient to study how to approximate optimal Q-value functions.
>
> **A1**: Our work seeks to theoretically understand DQN in perspective of neural networks’ function class and their architecture. We follow the key characteristics of DQN that works in the online setting and uses $\epsilon$-greedy for exploration.
>
> In fact, we have clarified that, when using  $\epsilon$-greedy, our PSE assumption is satisfied for exploration, see the remark behind Assumption 1, see line 219.
>
> Even though the exploration issue in the online setting is solved, the $\epsilon$-greedy scheme is not enough to ensure the temporal difference  (TD) error is smaller than zero, failing in "optimism in the face of uncertainty".
> Accordingly, one key issue left: how to achieve sublinear regret when the temporal difference error exists, see the response to **Q5** for details as below.
>
> If we put the above discussion aside, we only compare with [1] in terms of the approximation ability. When compared to [1] on the Hölder space of target Q-function, the considered function class in our work is larger and more suitable than the Hölder space for neural networks. For example, the Barron space in this work is the “largest” function space represented by two-layer neural networks, refer to Appendix A (line 579-583), otherwise curse of dimensionality occurs. The considered Besov spaces are more general than the Hölder space for deep ReLU neural networks for function approximation, refer to [S1] for details.
>
> More importantly, based on our theoretical results, we provide width-depth guidelines for understanding DQN according to the smoothness of the target Q function, which also coincide with the practical DQN.
>
> ---
>
> **Q2:** To my knowledge, the value iteration algorithm (with neural networks) is rarely used in practice, meaning that a gap separates the theory in this paper from practical algorithms.
>
> **A2:** To us the main difference is that we assume an exact solution at step 6 of Algorithm 1 while DQN takes some gradient steps of this objective.  Also DQN estimates the Bellman squared error with historical data so this does not seem to be a big difference.
>
> ---
>
> **Q3:** it is hard to compute the global minimizer defined in the empirical objective (i.e., Equation (3)) for each iteration. The computation cost is unacceptable.
>
> **A3:** We agree with the reviewer that in practice the computation cost is large. We follow the setting of [S1,S2,S3] in deep learning theory that directly assumes the global minima can be found. In particular, at least for two-layer neural networks beyond NTK, all minima can be obtained by  convex optimization algorithms [S4]. Besides, in deep learning theory, one global minimum can be also found via over-parameterization [S5]. We will mention this point in our revision.
>
> Note that in the case of approximate solutions for linear/kernel MDP conducted by gradient descent (as well as SGD) after $K$ steps, we do have a clear solution. Indeed, we can modify the line 7 in Alg.1 by introducing the extra  bonus function:
> $\widehat{Q}_h^t = \min \Big( Q_h^t + (1-\eta \lambda)^{K}HT, H-h+1  \Big)^{+} $
> where $\eta$ denotes the step-size satisfying $\eta < 1/\lambda$.
>
> Accordingly, such an approximation solution incurs an extra residual term $\mathcal{O}((1-\eta \lambda)^{K} HT)$ in the regret bound that exponentially converges to zero.

---

> > ### Author Response · Authors · 2022-08-01
> > **Reviewer Pwx6 (Part II)**
> >
> > **Q4:** Q-learning with function approximation can easily diverge, and the target network is introduced to address this issue in DQN. However, these features are not captured by Algorithm 1 and theoretical results.
> >
> > **A4:** We agree with the reviewer that one reason for the success of DQN in practice is due to the target neural networks, but understanding DQN from another perspective, i.e., function space and architectures (e.g., width and depth) deserves some attention.
> >
> > Interestingly, removing the target neural networks in DQN is also possible and acheives good performance in practice [S6].
> > This is similar to the scenario that training neural networks by SGD sometimes is difficult to converge, and we use some techniques, e.g., batch normalization, for training. But this does not mean, theoretical understanding of SGD without batch normalization is not acceptable.
> >
> > Moreover, we thank the reviewer for providing a relevant paper [5] on understanding DQN with the target neural network and linear function approximation, though linear function approximation could be far away from understanding practical DQN.
> > In fact, understanding DQN via the target neural networks and function space are mutually complementary rather than mutually exclusive. Both of them are important but could not have the right to exclude all others.
> >
> > **Q5:** Can the authors justify the (technical) challenge after Assumption 1 is satisfied?
> >
> > **A5:** The goal of Assumption 1 is to transform the TD error in $L_{\infty}$ space to $L_2(\mathrm{d} \mu)$ space. Even if Assumption 1 is used, the key technical challenge is how to bound the temporal difference (TD) error.
> >
> > To be specific, previous work on linear/kernel function approximation [S6,S7] design a UCB-type bonus function to ensure the temporal difference error is smaller than zero. However, this UCB-type bonus function depends on the explicit feature mapping that is definitely unknown for neural networks.
> >
> > In our case, we have to bound the TD error without such bonus function design. Using Assumption 1, the TD error in $L_{\infty}$-norm can be transformed to $L_2(\mathrm{d} \mu)$ space.
> > Accordingly, our key technical challenge is to transform the TD error estimation to generalization bounds under the non-iid data, see line 319 to 334 for details.
> >
> > **Q6:** Assumption 1 is quite strong for studying online learning. Though this paper explains that epsilon-greedy can achieve this, it is known that epsilon-greedy is much inefficient.
> >
> > **A6:** We agree with the reviewer that epsilon-greedy is inefficient under some worst cases according to Jin et al. (2018). However,  epsilon-greedy is still used in DQN in practice. Besides, in theory, epsilon-greedy is theoretically demonstrated to be efficient under some “well-structured” MDP [S9].
> >
> > -------refs-------
> >
> > [S1] Suzuki, Taiji. Adaptivity of deep ReLU network for learning in Besov and mixed smooth Besov spaces: optimal rate and curse of dimensionality. ICLR2019.
> >
> > [S2] Schmidt-Hieber, Johannes. Nonparametric regression using deep neural networks with ReLU activation function. *The Annals of Statistics*, 2020.
> >
> > [S3] Lin, Shao-Bo, Kaidong Wang, Yao Wang, and Ding-Xuan Zhou. Universal consistency of deep convolutional neural networks. *IEEE Transactions on Information Theory*, 2022.
> >
> > [S4] Wang, Yifei, Jonathan Lacotte, and Mert Pilanci. The Hidden Convex Optimization Landscape of Two-Layer ReLU Neural Networks: an Exact Characterization of the Optimal Solutions. ICLR2022.
> >
> > [S5] Allen-Zhu, Zeyuan, Yuanzhi Li, and Zhao Song. A convergence theory for deep learning via over-parameterization. ICML2019.
> >
> > [S6] Chi Jin, Zhuoran Yang, Zhaoran Wang, and Michael I Jordan. Provably efficient reinforcement learning with linear function approximation. COLT2020.
> >
> > [S7] Zhuoran Yang, Chi Jin, Zhaoran Wang, Mengdi Wang, and Michael I Jordan. On function approximation in reinforcement learning: Optimism in the face of large state spaces. NeurIPS2020.
> >
> > [S8] Seungchan Kim, Kavosh Asadi, Michael Littman, and George Konidaris. Deepmellow: removing the need for a target network in deep Q-learning. IJCAI2019.
> >
> > [S9] Dann, C., Mansour, Y., Mohri, M., Sekhari, A., Sridharan, K. Guarantees for Epsilon-Greedy Reinforcement Learning with Function Approximation. ICML2022.

---

> > ### Comment · Reviewer_Pwx6 · 2022-08-05
> > **Thanks for Your Expalanation**
> >
> > Thanks for your detailed explanation. I have a better sense of the technical challenge mentioned in Q5 and A5, but I am more confused about the other parts. Let us discuss some major issues.  I would appreciate it if the authors could help me.
> >
> > **On Assumption 1 and Comparison with [12]**: You mention that the epsilon-greedy can satisfy Assumption 1, upon which the sublinear regret in Theorem 1 is achieved. Could you explain more about how to set epsilon-greedy to satisfy Assumption 1?  For me, this assumption seems to depend on the function class and the parameter $\epsilon$. Furthermore, it should hold uniformly for $\epsilon \geq 1/\sqrt{T}$.  If this assumption is well justified, I can accept the sublinear regret in Theorem 1.
> >
> > Also, in the Remark of Theorem 1, the width parameter in [12] is compared. However, if my understanding is correct, this paper and [12] focus on different settings (i.e., there is no Assumption 1 in [12]). Is this comparison fair?
> >
> >
> >
> > **On Target Network**: You explain that the difference between Algorithm 1 and the practical DQN algorithm is that DQN takes stochastic gradient steps to solve the empirical objective in Equation (3). Thus, your thought is that this difference is minor if the optimization error is well-controlled. If I misunderstand your claim, please correct me. But I have a different opinion. For DQN with the target network, the learning target does not change during a period, but the exploration continues. It seems that this practical scenario cannot be reduced to the framework in this paper. Hence, I much prefer the study in [5] by Zanette et al.
> >
> > Again, I understand your motivation, as explained in A4.  But, my concern is that Algorithm 1 may be quite different from the practical algorithm DQN. Algorithm 1 might be a new algorithm that can converge and be sample-efficient in the online setting. To my understanding, there is no sufficient evidence that Algorithm 1 is close to DQN.  DQN is a simple online algorithm that can handle streaming data (so the computation costs could be cheap per iteration). In my view, DQN is different from the VI-style algorithms (VI means value iteration) in this paper and [S6], which are much similar to model-based algorithms. To my best knowledge, only [5] is close to the goal of designing sample-efficient algorithms with low computation costs. If differences between Algorithm 1 and DQN are not negligible, let us accept and clarify these differences.

---

> > > ### Author Response · Authors · 2022-08-05
> > > **Response to Reviewer Pwx6 on epsilon-greedy and target networks**
> > >
> > > We are thankful to the Reviewer Pwx6 with the constructive feedback.
> > >
> > >
> > > **Q1:** how to set epsilon-greedy to satisfy Assumption 1? For me, this assumption seems to depend on the function class and the parameter $\delta$
> > >  (**we change the parameter as $\delta$ here to avoid misunderstanding with $\epsilon$-greedy**).
> > >
> > > **A1:**
> > > Assumption 1 is used to build the relationship between $L^2(\mathrm{d} \bar{\mu}_h^t) $ and  $L^{\infty}$ norm.
> > >
> > > In our proof, we only require Assumption 1 holds for $f$ equivalent to the temporal difference error $\Gamma_h^t$, refer to Lemma 3 (line 676) for details.
> > >  Moreover, we add a proof  that the $\epsilon$-greedy policy satisfies Assumption 1 for any $\epsilon \in (0,1)$, refer to Appendix I (line 1000) for proofs in the revised version.
> > >
> > > In particular, we have (in the worst case with $\delta$ is zero)
> > > $$
> > >         \bar{\mu}^t_h(\mathcal{G}_{\delta}) \geqslant \Omega \left( \Big( \frac{\epsilon}{A} \Big)^H \right) > 0, \quad \forall \epsilon \in (0,1)~ and~~ \delta \geqslant 0 .$$
> > >
> > > Further, the exponential dependence on $H$ (in the worst case for any MDP under $\epsilon$-greedy) can be avoided at an additional cost of worsening $T$ dependence. For example,  in the Besov space, by setting $ \epsilon = \mathcal{O}(H^{\frac{4}{H+2}} T^{-\frac{1}{ (H+2) }}) $, we can recover Theorem 3 [S9]’s sublinear regret bound $\widetilde{\mathcal{O}} (H T^{\frac{H+1}{H+2}})$ in the worst case.
> > >
> > > ---
> > >
> > > **Q2:** fair comparison with [12]
> > >
> > > **A2:** As written in line 254 in the previous version, we believe
> > >
> > > “The results should be compared with care because the focus of the two works is different.”
> > >
> > > Albeit not directly comparable, we mentioned this work because this is the closest literature to our scope.
> > > We can remove this comparison in the final version if this is misleading in your view.
> > >
> > > ---
> > >
> > > **Q3:** discussion on target networks
> > >
> > > **A3:** Thank you for following up on this point. We agree with the reviewer's claim that even if the optimization error is neglected, then our algorithm becomes close to DQN without target networks (studied in [S6]) but not to the most famous version of DQN that uses a target network.
> > >
> > > We opted for this name because our algorithm 1 is a value based method using deep neural network function approximation that shares the same spirit and main intention of DQN. However, we really do not want to create confusion in the community and we acknowledge the fact that the study of target neural networks is beyond the scope of this work.
> > >
> > > To this end, we have renamed our algorithm “**Online Fitted Q Iteration with Deep Neural Networks**” in the revised version. We would welcome suggestions from the reviewer’s side about this renaming for Algorithm 1. Feel free to suggest any alternative that sounds more appropriate in your view.
> > >
> > > Thanks again for your constructive and helpful review!
> > >
> > > Best,
> > >
> > > Authors.

---

> > > > ### Comment · Reviewer_Pwx6 · 2022-08-07
> > > > **Increase My Score to 6**
> > > >
> > > > Thanks for clarifying these parts. My concerns are well addressed. Accordingly, I have changed my score to 6.

---

> > > > > ### Author Response · Authors · 2022-08-07
> > > > > **thanks for your positive support**
> > > > >
> > > > > Dear Reviewer Pwx6,
> > > > >
> > > > > Thanks for your positive support.
> > > > >
> > > > > We will continue polishing our work based on your constructive feedback and suggestions in the final version.
> > > > >
> > > > > Best regards,
> > > > >
> > > > > Authors.

---

### Official Review · Reviewer_7U3N · 2022-07-11

**Rating:** 6
**Confidence:** 4
**Soundness:** 2 fair
**Presentation:** 3 good
**Contribution:** 2 fair

**Summary:**

This paper theoretically studies value iteration type algorithms in RL using neural networks belonging to either the Besov or Barron function spaces. Under the probabilistic sufficient exploration assumption, scalings of the width and depth of the neural network are revealed for achieving sublinear regret for the proposed least square value iteration with neural networks. It is argued that the width has a more prominent effect than depth. Quantitative regret bound is established and only depends logarithmically on the dimension $d$.

**Questions:**

Some comments and questions are as follows:

1. There is a mismatch between the terminology 'Q-learning' and the actual algorithm studied in this paper, which is indeed a least square value iteration algorithm with a neural network estimator. This is different from the usual Q-learning type algorithm, where the Q updates correspond to gradient descent of one-step TD error. This should be clarified in the paper.
2. The regret bounds in Theorem 1 and 2 scale with $T^{4/3}$ and $T^{\frac{\alpha+d}{2\alpha+d}}$, which are strictly worse than $\sqrt{T}$. Do these results degenerate to $\sqrt{T}$-type bound under simpler settings such as linear function approximation?
3. Why there is no need to do exploration in Algorithm 1? It should be necessary for online RL algorithms. Is this due to the PSE assumption?
4. Indeed, the PSE assumption seems unclear to me. In the definition of $\mathcal{G}_\epsilon$, is $f$ an arbitrary function? Here $\epsilon$ seems to depend on the value of $\|f\|_\infty$, and it is unclear where $f$ is specified. Apart from this, by letting $\bar\mu_h^t(\mathcal{G}_\epsilon) > 0$ hold with probability at least $1-\delta$, it is imposing some condition on $\widehat Q_h^1, \ldots, \widehat Q_h^T$, which is obtained in an online manner. Then, conditioning on Assumption 1 being true, $\{\widehat Q_h^t\}$ is not a martingale sequence any more as conditioning changes its distribution (so do the TD error), thus it needs to be justified how to use martingale concentration inequalities (line 307-308).
5. When running Algorithm 1, the collected data is conditionally independent instead of independent, because the policy in episode $t$ depends on all the previous data. Then how can we apply Lemma 5/6 (line 320-321)?

**Limitations:**

The authors have adequately addressed the limitations and potential negative societal impact of their work.

**Strengths And Weaknesses:**

This paper is easy to follow, and it is an important topic to analyze RL algorithms that employ neural network type estimators. The algorithm studied in this paper (Algorithm 1) is well-known, while it is claimed that the theoretical analysis reveals guidelines for the training setup in real applications. The discussion about related works for RL with function approximation can be more detailed, especially comparing the difference between the assumptions on the function classes. Also, it seems that some existing literature on theoretical analysis of Q-learning algorithms is missing, e.g., [1] and [2].



Reference:

[1] Chi Jin, Zeyuan Allen-Zhu, Sebastien Bubeck, Michael I. Jordan. Is Q-learning Provably Efficient?

[2] Pan Xu, Quanquan Gu. A finite-time analysis of Q-learning with neural network function approximation.

---

> ### Author Response · Authors · 2022-08-01
> **Response to Reviewer 7U3N (part I)**
>
> We thank the reviewer for the effort and the constructive feedback!
>
> **Q1.** The discussion about related works for RL with function approximation can be more detailed, especially comparing the difference between the assumptions on the function classes. Also, it seems that some existing literature on *theoretical analysis of Q-learning algorithms is missing*.
>
> **A1:** Thanks for pointing out two relevant refs on Q-learning. We have added them in the revised version and briefly discuss them here: Jin et al. (2018) prove that Q-learning with UCB exploration achieves $\sqrt{T}-$regret in tabular MDPs; Pan and Gu (2020) shows that neural Q-learning (under the lazy training regime) finds the optimal policy with $\mathcal{O}(1/T)$ convergence rate. Both of these two refs do not weaken the contribution of this work.
> We will add more discussion on these work according to the used assumption, the function class, and the obtained regret in the final version.
> ---
>
> **Q2:** There is a mismatch between the terminology 'Q-learning' and the actual algorithm studied in this paper, which is indeed a least square value iteration algorithm with a neural network estimator. This is different from the usual Q-learning type algorithm, where the Q updates correspond to gradient descent of one-step TD error. This should be clarified in the paper.
>
> **A2:**  Thank you for pointing this out. $Q$-learning is indeed a gradient descent of one-step TD error. According to your suggestions, we have adopted the terminology centered around value iteration in the revised manuscript to make it more precise.  For example, the title in section 4 is changed to “Theoretical guarantees of efficient value iteration via DNNs”.
>
> The title will be also changed to “Width and Depth Guidelines for DQN via Value Iteration: A Function Approximation Perspective”.
>
> ---
>
> **Q3:** The regret bounds in Theorem 1 and 2 scale with $T^{-¾}$ and $T^{- \frac{\alpha + d}{2 \alpha + d}}$ which are strictly worse than $T$. Do these results degenerate to T-type bound under simpler settings such as linear function approximation?
>
> **A3:** Our result can obtain $\sqrt{T}$-regret under linear/kernel function approximation. To be specific, the key step in our proof is transforming the estimation on temporal difference error to generalization bounds on non-iid data. In Theorem 1, the optimal generalization rate of learning in Barron space is $\mathcal{O}(1/\sqrt{n})$, where $n$ is the number of training data (i.e., $t-1$ in our RL setting). If we consider the linear/kernel function class, the generalization rate can be improved to $\mathcal{O}(1/n)$, which obtains a $\sqrt{T}$-regret bound. The similar discussion is suitable to Theorem 2.
> Furthermore, we provide another view to understand this. In our theorem 2, the parameter $\alpha$ denotes the smoothness of the $Q^{\star}$. Since the linear function is infinitely smooth, i.e., $\alpha \rightarrow \infty$, our regret is arbitrarily close to $O(1/\sqrt{T})$.
>
> ---
>
> **Q4:** Why there is no need to do exploration in Algorithm 1? It should be necessary for online RL algorithms. Is this due to the PSE assumption?
>
> **A4:** Exactly. Our assumption requires that at least one good state-action pair can be visited according to the historical data. Note that exploration in DQN depends on massive data and $\epsilon$-greedy scheme rather than designing UCB-type bonus function, and more importantly, the used **$\epsilon$-greedy scheme satisfies our PSE assumption**, which coincides with DQN in practice.
> Therefore, we decided to study the properties of the DQN architecture leaving aside the design of an exploration scheme for nonlinear function approximation. In fact, **designing UCB-style exploration in neural function approximation beyond NTK is still an open question** as no explicit feature mapping can be given.
> We have already discussed this in the remark behind Assumption 1, see line 219 for details.

---

> > ### Author Response · Authors · 2022-08-01
> > **Response to Reviewer 7U3N (part II)**
> >
> > **Q5:** about PSE assumption and $\widehat{Q}_h^t$ is not a martingale sequence
> >
> > **A5:** Here the function is restricted to $f \in \mathcal{F}$ which is $ L^p_{d \mu}$-integrable, where $\mu$ is the averaged measure  over the historial state-action pairs, and $\epsilon \leq || f ||_{\infty}$ ensures the RHS of Eq. (7) nonnegative.
> > In our setting, $f$ representing the temporal difference error is enough for proof.
> >
> > When employing $\epsilon$-greedy exploration, our PSE assumption satisfied. In this case,the martingale sequence is not needed, instead it incurs in extra $\epsilon H T$ regret in the regret bound. By taking $\epsilon = T^{-\alpha}$ for some certain $\alpha \in (0,1)$, the sublinear regret still holds under the $\epsilon$-greedy policy.
> >
> > Besides, for deterministic dynamics in discrete states and actions spaces, we can use the first $|\mathcal{S}| | \mathcal{A}|/H$ episodes to visit each state-action pair once, which is feasible in Atari, and accordingly our PSE assumption is satisfied. In this case, the martingale structure still remains unchanged.
> >
> > ---
> >
> > **Q6:** When running Algorithm 1, the collected data is **conditionally independent** instead of independent, because the policy in episode t depends on all the previous data. Then how can we apply Lemma 5/6 (line 320-321)?
> >
> > **A6:** True, we did not notice this point. Thanks a lot for noticing it!
> >
> > Following  a covering argument for the function approximation class used for value functions [S1], **we have solved the issue**, refer to Appendix H in the revised version for details. We would sincerely appreciate it if the reviewer could check it.
> >
> > Briefly speaking, we propose to act greedly not with respect to $\hat{Q}^t_h$ but **with respect to the closest deterministic elements of the covering set**. In this way we break the dependence between the episodes, introduce intermediate random variables based on independent data for unbiased estimation, and accordingly Lemmas 5 and 6 can be applied.
> >
> > By doing so, two extra terms occur: one is the expected risk difference between independent and conditionally independent data; and the other is caused by a biased estimator. Both of these two terms can be estimated by the $\epsilon$-covering. By taking $\epsilon = H/ \sqrt{t}$ following [S1], the final regret bound still remains unchanged.
> > Besides, in practice, such independence requirements can be achieved by the experience replay in DQN.
> >
> > We are grateful to the reviewer for pointing out this issue and would be glad to elaborate if the reviewer has further concerns.
> >
> >
> > [S1] Jin, Chi, Zhuoran Yang, Zhaoran Wang, and Michael I. Jordan. Provably efficient reinforcement learning with linear function approximation. COLT2020.

---

> > > ### Comment · Reviewer_7U3N · 2022-08-06
> > > **Thanks for the clarifications**
> > >
> > > First I thank the authors for the explanations. Yet I still have some concerns as follows:
> > >
> > > 1. I have read the reviews from other reviewers and the corresponding responses. Especially, I agree with Reviewer Pwx6's concern that there is a big discrepancy between Algorithm 1 and the standard DQN algorithm, as I have also mentioned in the review. I think they are different even without experience replay. Line 6 of Algorithm 1, or equivalently minimizing Eq. (3), is basically solving the problem of finding a fixed point of the TD error based on the collected data, which is essentially very different from _one_ step of (mini-batch) gradient descent. However, most of the motivation part of this paper actually discusses the practical DQN, which I think is misleading.
> > > 2. Regarding the PSE assumption, the authors have added a proof that using $\epsilon$-greedy policy guarantees PSE. I agree that in this case there is no issue about $\widehat Q_h^1, \ldots, \widehat Q_h^T$ being a martingale sequence. However, as long as the PSE assumption (implicitly) affects the distribution of $\widehat Q_h^1, \ldots, \widehat Q_h^T$, it is no longer a martingale sequence. I feel it is hard to examine this in general.
> > > 3. Also thanks for the clarifications on the issue of conditional independence. According to the response, the authors proposed to act greedily with respect to the closest deterministic elements of the covering net. Then my question is how to find the closest deterministic element? Meanwhile, the size of the $\epsilon$-net should be exponential in dimension $d$ (at least for the linear case as far as I know). Is this even implementable?

---

> > > > ### Author Response · Authors · 2022-08-07
> > > > **Response to Reviewer 7U3N on DQN, $\epsilon$-greedy, and covering number**
> > > >
> > > > We are thankful to the Reviewer 7U3N’s constructive and detailed feedback!
> > > >
> > > > We agree with both two reviewers’ suggestion on the difference with DQN, and accordingly we have restated the story of this work and renamed the title to " **Understanding Deep Neural Function Approximation in Reinforcement Learning with $\epsilon$-Greedy** ".
> > > >
> > > > ---
> > > >
> > > > **Q1:** difference with DQN
> > > >
> > > > **A1:** We agree with the reviewer that our algorithm based on value iteration via deep neural networks is different from one-step gradient descent in Q learning.
> > > > Theoretically understanding DQN can be conducted in the perspective of function approximation, target network, and Q-learning.
> > > >
> > > > Our motivation focuses on deep neural function approximation beyond "linear" regime that shares the same spirit and main intention of DQN, and thus provides some guidelines on architecture of neural networks (width and depth) to ensure sublinear regret based on different smoothness levels of the target Q-function.
> > > >
> > > > We really do not want to create confusion in the community and we acknowledge the difference between DQN and Algorithm 1, and thus we have renamed our algorithm “ **Value Iteration via neural networks under $\epsilon$-greedy** ”. We have revised this paper to avoid misleading, refer to [paper revision response](https://openreview.net/forum?id=o8vYKDWMnq1&noteId=qlRlvydlhYM9).
> > > >
> > > > ---
> > > >
> > > > **Q2:** PSE assumption and martingale sequences
> > > >
> > > > **A2:** We agree with the reviewer that, it is hard to examine whether our PSE assumption (implicitly) affects the distribution of $\widehat{Q}_h^1, \widehat{Q}_h^2, \cdots, \widehat{Q}_h^t$.
> > > >
> > > > According to your suggestions, we remove our PSE assumption but use the $\epsilon$-greedy policy in Algorithm 1. In this case, this issue has been solved. Interestingly, understanding deep neural function approximation with $\epsilon$-greedy is still promising in the RL theory community, and even more close to DQN that also uses $\epsilon$-greedy in practice.
> > > >
> > > >  Accordingly, we follow this story to restate our motivation, rename the title to “ **Understanding Deep Neural Function Approximation in Reinforcement Learning with $\epsilon$-Greedy** ”, and provide a possible way to understand deep RL in terms of function class under the $\epsilon$-greedy exploration. We have tried our best to avoid misleading this work, refer to the revised version for details.
> > > >
> > > > ---
> > > >
> > > > **Q3:** how to find the closest deterministic element? And the size of $\epsilon$-net is exponential order of $d$. Is this even implementable?
> > > >
> > > > **A3:** In Eq. (45), line 979, we introduce $\widetilde{Q}_h^t$ as a proxy for $\widehat{Q}_h^t$ and we account for the mismatch between the policy greedy w.r.t. $\widetilde{Q}$ and w.r.t. $\widehat{Q}$ in the analysis. We don’t need to find such a point as $\widetilde{Q}_h^t$ is an intermediate random variable to aid our proof.
> > > > Besides, in Jin et al. (COLT2020) for linear function approximation, they also don’t need to find the closest deterministic element, see the proof of Lemma D.4 in Jin et al. (COLT2020).
> > > >
> > > > It’s true that the size of $\epsilon$-net is exponential order of $d$, but what we use is the logarithm of covering number, i.e., metric entropy, in our proof. This is also conducted by Lemma D.4 and D.6. in Jin et al. (COLT2020) via this style.
> > > >
> > > > ---
> > > > [Jin et al. (COLT2020)](https://arxiv.org/abs/1907.05388) Jin, Chi, Zhuoran Yang, Zhaoran Wang, and Michael I. Jordan. Provably efficient reinforcement learning with linear function approximation. COLT2020.
> > > >
> > > > We are happy to elaborate if the reviewer 7U3N has further questions!

---

> > > > > ### Comment · Reviewer_7U3N · 2022-08-08
> > > > > **Discussion with the authors**
> > > > >
> > > > > I thank the authors for their efforts.
> > > > >
> > > > > With these modifications, the results now make sense to me, and I will adjust my rating.

---

> > > > > > ### Author Response · Authors · 2022-08-08
> > > > > > **thanks for your positive support**
> > > > > >
> > > > > > Dear Reviewer 7U3N,
> > > > > >
> > > > > > We are grateful for your constructive feedback on improving this work.
> > > > > > We will polish this paper based on your suggestions in the final version.
> > > > > >
> > > > > > Best regards,
> > > > > >
> > > > > > Authors

---

### Official Review · Reviewer_TT8h · 2022-07-19

**Rating:** 7
**Confidence:** 4
**Soundness:** 3 good
**Presentation:** 3 good
**Contribution:** 3 good

**Summary:**

This paper provides a theoretical study of the successful deep Q-networks (DQN) framework under an online, episodic Markov decision process (MDP) model with T episodes. They focus on Besov and Barron function spaces in approximating an $\alpha$ smooth Q-function in a d-dimensional feature space. Under some assumptions, they prove that DQN with shallow layers and medium width can learn the MDP with sublinear regret.

**Questions:**

1. In practice, how large is the norm parameter B?
2. I do not understand why the regret bound in Theorem 1 not depend on dimension $d$ but in Theorem 2 it depends on $d$ explicitly. What is the reason behind such difference?
3. What is the role of sparsity parameter $S$ defined in Eq. (5)?


**Limitations:**

The authors have adequately addressed the limitations and potential negative societal impact of their work.

**Strengths And Weaknesses:**

Strengths:
In summary, this is a good paper for RL theory. It tries to give a theoretical understanding of the widely used DQN algorithm, which is fundamental in Deep RL. The theoretical results in this paper is also interesting. It shows that DQN-type algorithm can achieve sublinear regret using a small-scale neural network, under some assumptions. As far as I know, these results are novel in RL theory. Also, the results match the practical experience.

Weaknesses:
1. The algorithm requires optimizing a neural network function to minimum. However in practice, it is usually not achievable. What will the results be if we only have an approximate solution to the optimization problem?

---

> ### Author Response · Authors · 2022-08-01
> **Response to Reviewer TT8h**
>
> We thank the reviewer for constructive comments and positive support!
>
> **Q1:** The algorithm requires optimizing a neural network function to minimum. However in practice, it is usually not achievable. What will the results be if we only have an approximate solution to the optimization problem?
>
> **A1:** We agree with the reviewer that in practice it appears difficult to obtain the minimum solution. We follow the setting of [S1,S2,S3] in deep learning theory that directly assumes the global minima can be found. In particular, at least for two-layer neural networks beyond NTK, all minima can be obtained by  convex optimization algorithms [S4]. Besides, in deep learning theory, one global minimum can be also found via over-parameterization [S5]. We will mention this point in our revision.
>
> Note that in the case of approximate solutions for linear/kernel MDP conducted by gradient descent (as well as SGD) after $K$ steps, we do have a clear solution. Indeed, we can modify the line 7 in Alg.1 by introducing the extra  bonus function:
> $\widehat{Q}_h^t = \min \Big( Q_h^t + (1-\eta \lambda)^{K}HT, H-h+1  \Big)^{+} $
> where $\eta$ denotes the step-size satisfying $\eta < 1/\lambda$.
>
> Accordingly, such an approximation solution incurs an extra residual term $\mathcal{O}((1-\eta \lambda)^{K} HT)$ in the regret bound that exponentially converges to zero.
>
>
> **Q2:** In practice, how large is the norm parameter B?
>
> **A2:** Here we conducted one experiment of training an 8-layer fully-connected neural networks (without bias) under different widths (64, 256, 1024) as below. It can be found that, the norm parameter $B$ is smaller than 1 in practice.
>
> |Width | 64 | 256 | 1024 |
>
> |B | 0.845 | 0.424  | 0.249 |
>
> **Q3:** I do not understand why the regret bound in Theorem 1 not depend on dimension $d$ but in Theorem 2 it depends on $d$ explicitly. What is the reason behind such difference?
>
> **A3:** This is because the regret bound depends on generalization results in different function spaces, which admits different approximation ability, see line 313-316 for details.
>
> In Barron space, the approximation error converges at $\mathcal{O}(1/m)$ [S6], independent of $d$; while in the Besov space, the approximation rate is $\mathcal{O}(n^{-\frac{2\alpha}{2 \alpha + d}})$ [S1], that depends on $d$.
>
> **Q4:** What is the role of sparsity parameter $S$ defined in Eq. (5)?
>
> **A4:** This sparsity parameter $S$ depends on the fact that DNN with ReLU has few activations [S7], and further is used to control the capacity of the Besov spaces, see Appendix G for details.
>
>
> [S1] Suzuki, Taiji. Adaptivity of deep ReLU network for learning in Besov and mixed smooth Besov spaces: optimal rate and curse of dimensionality. ICLR2019.
>
> [S2] Schmidt-Hieber, Johannes. Nonparametric regression using deep neural networks with ReLU activation function. *The Annals of Statistics*, 2020.
>
> [S3] Lin, Shao-Bo, Kaidong Wang, Yao Wang, and Ding-Xuan Zhou. Universal consistency of deep convolutional neural networks. *IEEE Transactions on Information Theory*, 2022.
>
> [S4] Wang, Yifei, Jonathan Lacotte, and Mert Pilanci. The Hidden Convex Optimization Landscape of Two-Layer ReLU Neural Networks: an Exact Characterization of the Optimal Solutions. ICLR2022.
>
> [S5] Allen-Zhu, Zeyuan, Yuanzhi Li, and Zhao Song. A convergence theory for deep learning via over-parameterization. ICML2019.
>
> [S6] Weinan E, Chao Ma, and Lei Wu. The barron space and the flow-induced function spaces for neural network models. *Constructive Approximation*. 2021.
>
> [S7] Hanin, Boris, and David Rolnick. Deep ReLU networks have surprisingly few activation patterns. NeurIPS2019.

---

### Author Response · Authors · 2022-08-07
**General response on the paper revision**

We first thank all the reviewers' effort and constructive feedback on this work!

According to suggestions from all the reviewers, we have revised this paper to avoid misleading. We made our work center around **deep neural function approximation beyond the “linear” regime under $\epsilon$-greedy exploration in deep RL, affected by function class and architectures (e.g., width and depth)**. The main revision includes the following folds:

- Since the $\epsilon$-greedy exploration satisfies Assumption 1, with the proof in Lemma 2 in Appendix C, see line 649, we remove Assumption 1 but use $\epsilon$-greedy in the current version.
- We rename our Algorithm 1 “Value Iteration via neural networks under $\epsilon$-greedy exploration” as well as the paper title “ **Understanding Deep Neural Function Approximation in Reinforcement Learning with $\epsilon$-Greedy** ”  to avoid misleading DQN. In this case, we rephrase our expression on the relationship between our work and DQN in the abstract and the introduction. Moreover, a clarification is also given in the conclusion, refer to line 338 for details.
- Our regret bound is slightly changed as  $\epsilon$-greedy is applied. Specifically, in the special case (linear function approximation) of our work, our result is able to recover the optimal regret bound $\widetilde{\mathcal{O}}(H^{4/3} A^{1/3} T^{2/3})$ for contextual bandits under $\epsilon$-greedy [1], see line 249 for details.


Feel free to leave any comments and we are happy to hear your further suggestions on the revised version.

Best regards,

Authors

---
[1] Lattimore, Tor, and Csaba Szepesvári. Bandit algorithms. Cambridge University Press, 2020.

---

### Meta-Review · Area_Chair_Dyv4 · 2022-08-26

**Recommendation:** Accept
**Confidence:** Certain

**Metareview:**

This paper provides a theoretical study of the successful deep Q-networks framework under an online, episodic Markov decision process (MDP) model with T episodes. All reviewers and the AC believe this is a solid RL theory paper.

**Award:**

No

---

### Decision · Program_Chairs · 2022-09-14

Accept